



# Insights into aerosol chemistry during the 2015 China victory day parade: results from simultaneous measurements at ground level and 260 m in Beijing

Jian Zhao[1,3], Wei Du[1,3], Yingjie Zhang[4,1], Qingqing Wang[1], Chen Chen[1], Weiqi Xu[1,3], Tingting Han[1,3], Yuying Wang[5], Pingqing Fu[1], Zifa Wang[1], Zhanqing Li[5], Yele Sun[1,2*]

[1]State Key Laboratory of Atmospheric Boundary Layer Physics and Atmospheric Chemistry, Institute of Atmospheric Physics, Chinese Academy of Sciences, Beijing 100029, China

[2]Center for Excellence in Urban Atmospheric Environment, Institute of Urban Environment, Chinese Academy of Sciences, Xiamen 361021, China

[3]College of Earth Sciences, University of Chinese Academy of Sciences, Beijing 100049, China

[4]Collaborative Innovation Center on Forecast and Evaluation of Meteorological Disasters, Nanjing University of Information Science & Technology, Nanjing 210044, China

[5]College of Global Change and Earth System Science, Beijing Normal University, Beijing 100875, China

*Correspondence to*: Y. L. Sun (sunyele@mail.iap.ac.cn)

**Abstract.** Strict emission controls were implemented in Beijing and adjacent provinces to ensure good air quality during the 2015 China victory day parade. Here we conducted synchronous measurements of submicron aerosols ($PM_1$) at ground level and 260 m on a meteorological tower by using a High-Resolution Aerosol Mass Spectrometer and an Aerosol Chemical Speciation Monitor, respectively, in Beijing from 22 August to 30 September. Our results showed that the average $PM_1$ concentrations are 19.3 and 14.8 µg m$^{-3}$ at ground level and 260 m, respectively, during the control period (20 August – 3 September), which are 57% and 50% lower than those after the control period (4 – 30 September). Organic aerosols (OA) dominated $PM_1$ during the control period at both ground level and 260 m (55% and 53%, respectively), while its contribution showed substantial decreases (~40%) associated with an increase in secondary inorganic aerosols (SIA) after the parade indicating a larger impact of emission controls on SIA than OA. Positive matrix factorization of OA further illustrated that primary OA (POA) showed similar decreases as secondary OA (SOA) at both ground level (40% vs. 42%) and 260 m (35% vs. 36%). However, we also observed significant changes in SOA composition. While the more oxidized SOA showed a large decrease by 75%, the less oxidized SOA was comparable during (5.6 µg m$^{-3}$) and after the control periods (6.5 µg m$^{-3}$). Our results demonstrated that the changes in meteorological conditions and PM loadings have affected SOA formation mechanisms, and the photochemical production of fresh SOA was more important during the control period. By isolating the influences of meteorological conditions and footprint regions in polluted episodes, we found that regional emission controls on average reduced PM levels by 44 – 45%, and the reductions were close among SIA, SOA and POA at 260 m, whereas primary species showed relatively more reductions (55 – 67%) than secondary aerosol species (33% - 44%) at ground level.





## 1 Introduction

Beijing, the capital of China with ~21.71 million people in the metropolitan area (Beijing Municipal Bureau of Statistics, 2015), has been suffering from severe air pollution during the past decade (Huang et al., 2014;Guo et al., 2014). According to Beijing Environmental Statement, the annual average mass concentration of fine particles ($PM_{2.5}$) is 80.6 µg m$^{-3}$ in Beijing

in 2015, which was decreased by 6.2% compared with that in 2014. However, the concentration was more than twice the China National Ambient Air Quality Standard (35 µg m$^{-3}$ as an annual average) suggesting that air pollution in Beijing is still severe. For these reasons, extensive studies utilizing various offline and online techniques have been conducted to investigate the characteristics of aerosol particles and the formation mechanisms of haze episodes (Zhao et al., 2013;Tian et al., 2014;Elser et al., 2016). Among them, Aerodyne Aerosol Mass Spectrometer (AMS) is unique which is capable of

getting size-resolved chemical information of non-refractory submicron aerosols (NR-$PM_1$) in real-time (DeCarlo et al., 2006;Canagaratna et al., 2007). While the research grade AMS has been widely used in air pollution studies in China (Sun et al., 2010;Huang et al., 2010;Huang et al., 2011;Xu et al., 2015;Hu et al., 2016b), the deployments of the compact and robust Aerosol Chemical Speciation Monitor (ACSM) (Ng et al., 2011) is relatively new (Sun et al., 2012;Zhang et al., 2015b;Li et al., 2016). Together, these studies have significantly improved our understanding of the composition, processes and sources

of submicron aerosols ($PM_1$). For example, long-term monitoring results showed that organics contributed the largest fraction of $PM_1$ (40-51%) during all seasons in urban Beijing, followed by nitrate (17-25%) and sulfate (12-17%) (Sun et al., 2015b). Positive matrix factorization analysis further highlighted substantial different organic aerosols (OA) composition in different seasons. Generally, secondary OA (SOA) is the major component of OA in summer, on average accounting for 57-64% (Huang et al., 2010;Sun et al., 2010;Sun et al., 2012;Zhang et al., 2015a), while primary OA (POA) (~70%) made a

higher contribution in winter and even as much as ~70%, especially with the significantly enhanced coal combustion OA in heating seasons (Liu et al., 2011;Sun et al., 2013b;Hu et al., 2016a). Back trajectory analysis showed that high $PM_1$ mass concentrations were often associated with air masses from the south and southwest, and the composition was characterized by high fractions of oxidized OA and secondary inorganic aerosols (SIA = sulfate + nitrate + ammonium), whereas primary species emitted from local sources were the dominant contributors during clean events (Huang et al., 2010;Sun et al.,

2010;Sun et al., 2012;Zhang et al., 2015a). In addition, the impacts of meteorological parameters (e.g. temperature, relative humidity) on the formation and evolution of pollution events in winter were also extensively studied (Sun et al., 2013a;Sun et al., 2013b;Zhang et al., 2014;Wang et al., 2015;Jiang et al., 2015) . Results showed that stagnant meteorological conditions, local emissions, regional transport and secondary production were four major factors leading to the formation of severe haze episodes in Beijing in winter (Sun et al., 2014;Zhang et al., 2014;Liu et al., 2016).

However, most previous studies were conducted at ground sites which are subject to strong influences from local sources, e.g., traffic and cooking emissions. The mixed local emissions increase the difficulties in quantifying the relative





contributions of local sources and regional transport to severe haze pollution. Despite this, real-time characterization of submicron aerosols at high altitudes in the megacity of Beijing is rare. Sun et al. (2015a) conducted the first real-time measurements of $PM_1$ at 260 m on a meteorological tower using an ACSM along with synchronous measurements at ground site by an Aerodyne High-resolution Time-of-Fight AMS (HR-AMS hereafter). While secondary aerosol species had similar

temporal variations between 260 m and ground site, the variations in primary species were substantially different. In addition, relatively higher contribution of nitrate was observed at 260 m (15%) than ground site (10%), which was likely due to the enhanced gas-particle partitioning associated with lower temperature and higher relative humidity (RH) at 260 m. Chen et al. (2015) further found that the variations in primary and secondary aerosol species were similar at 260 m, while they were significantly different at ground site. These results illustrated a significant impact of regional transport on aerosol species at

high altitudes. However, due to the limited studies at different heights in the city, our knowledge of the sources and evolutionary processes of aerosol particles is far from complete.

Understanding of the relationship between source emissions and aerosol chemistry is of great importance for mitigating air pollution in megacities. For example, strict emission controls were often implemented in Beijing and surrounding regions to ensure good air quality during specific events, e.g., Beijing Olympic Games in 2008 and Asia-Pacific Economic Cooperation

(APEC) summit in 2014, which offered experimental opportunities to study the impacts of source emission controls on air pollution. Huang et al. (2010) found that strict emission controls along with favorable meteorological conditions led to the low $PM_1$ levels during the Beijing Olympic Games. Similarly, all $PM_1$ species at 260 m were observed to decrease substantially (40-80%) as a response to strict emission controls over regional scale during APEC (Chen et al., 2015). In contrast, secondary species showed substantial reductions at ground level, ranging from 35% to 69%, yet the reductions in

primary species were much smaller (Xu et al., 2015). These studies highlight the importance of regional emission controls in decreasing secondary aerosol precursors, and hence suppressing the formation of secondary aerosols, yet a more strict local emission control is also needed. Despite this, our understanding of the relationship between aerosol chemistry and emission controls is not complete mainly due to the variations of aerosol composition and processes under different meteorological conditions.

To ensure good air quality during the 2015 China victory day parade, a series of temporary control measures were implemented in Beijing and surrounding regions from 20 August to 3 September. These strict emission controls included stopping construction activities, restricting the number of vehicles by alternating odd and even plate numbers, prohibiting outdoor barbecue, and shutting down factories and power plants in adjacent provinces. Such measures were even stricter than those during APEC by extending control areas to Shandong province. As a consequence, the air quality hit the best

record since the release of $PM_{2.5}$ standard in 2012, leading to 15 blue sky days, which was commonly referred to "Parade Blue".      The      average      $PM_{2.5}$      concentration      was      18      μg      $m^{-3}$      during      the      control      period



(http://www.gov.cn/xinwen/2015-09/07/content_2926447.htm), which was decreased by 73.1% compared to that during the same period in 2014. In addition, the PM levels and meteorological conditions were both significantly different from those during the 2014 APEC summit. Thus, this study period is unique in investigating aerosol characteristics during low PM level periods and further evaluating the impacts of emission controls on aerosol sources, composition and processes.

Here we deployed an HR-AMS and an ACSM at ground level and 260 m, respectively, for synchronous measurements of aerosol particle composition, including organics (Org), sulfate ($SO_4$), nitrate ($NO_3$), ammonium ($NH_4$), and chloride (Chl) from 22 August to 30 September, 2015, along with collocated measurements of black carbon (BC) and size-resolved number concentrations. The mass concentrations, chemical composition, and diurnal variations of submicron aerosol species at ground level and 260 m during and after the control period were characterized and compared. The sources of OA at different

heights were investigated by positive matrix factorization and the changes in OA composition due to emission controls were elucidated. In addition, the sources of aerosol particle species during and after the control period were investigated with FLEXPART analysis and bivariate polar analysis.

## 2 Experimental methods

### 2.1 Sampling site and meteorology

The sampling site is located at Tower branch of the Institute of Atmospheric Physics (IAP), CAS (39°58'28" N, 116°22'16" E, ASL: 49 m), which is a typical urban site that is subject to influences from local traffic and cooking emissions nearby (Sun et al., 2015b). The detailed information about this sampling site was presented in previous studies (Sun et al., 2012;Sun et al., 2015b). The meteorological parameters including temperature ($T$), RH, and winds (wind speed and wind direction) were obtained from the Beijing 325 m meteorological tower at 15 heights (8, 15, 32, 47, 65, 80, 100, 120, 140, 160, 180, 200, 240,

280 and 320 m). While the average RH, $T$, and wind speed were relatively similar between the control (20 August – 3 September) and non-control periods (4 September – 30 September) (Table 1), the prevailing winds were substantially different. The control period was characterized by dominant northerly winds while the non-control period showed prevailing winds from both the northerly and southerly (Fig. S1). All the data in this study was reported at ambient pressure and temperature in Beijing Time (i.e., Coordinated Universal Time + 8 h).

### 2.2 Instrumentation and operation

The HR-AMS and ACSM were deployed at ground level and 260 m, respectively, for synchronous measurement of NR-$PM_1$ species from 22 August to 30 September, 2015. The HR-AMS was situated in a sampling room located at the rooftop of a two-story building (~8 m). The ambient air was drawn into the sampling room through a stainless steel tube with the flow rate of 10 L min$^{-1}$, of which ~0.085 L min$^{-1}$ was isokinetically sampled into the HR-AMS. Coarse particles with aerodynamic



diameters larger than 2.5 μm were removed by a $PM_{2.5}$ URG cyclone (URG-2000-30ED) that was mounted in front of the sampling line. A silica gel diffusion dryer was also set up in front of the inlet system to avoid water vapor condensation which reduces the uncertainties in particle collection efficiency (CE) due to variable humidity (Matthew et al., 2008). Mass spectrum (MS) and particle time-of-flight (PToF) modes cycled every ~10 s in mass sensitive V-mode to obtain mass

concentrations and size distributions with a time resolution of 5 min. The high mass resolution W-mode (5 min) was also operated for subsequent high resolution spectral analysis and elemental analysis. The calibrations of HR-AMS were conducted following the standard protocols described in previous publications (Jayne et al., 2000;Jimenez, 2003).

Collocated instruments at the ground site included a two-wavelength Aethalometer (AE22, Magee Scientific Corp.) for BC, a cavity attenuated phase shift extinction monitor (CAPS $PM_{ext}$, Aerodyne Research Inc.) for light extinction (λ = 630

nm) of fine particles, and a CAPS $NO_2$ monitor (Aerodyne Research Inc.) for gaseous $NO_2$. Gas analyzers (Thermo Scientific) for $O_3$, CO, NO, and $NO_y$, and a tapered element oscillating microbalance (TEOM series 1400a, Thermo Scientific) for $PM_{2.5}$ were also deployed on the roof of an another two-story building, which is located approximately 30 m to the north. In addition, a tandem differential mobility analyzer (TDMA) that is developed by Guangzhou Institute of Tropical and Marine, China Meteorological Administration was used to measure size-resolved particle number

concentrations between 10 – 400 nm. A detailed description of the TDMA system during this study is presented elsewhere (Tan et al., 2013).

The ACSM and collocated instruments were stored in two containers at 260 m on the tower. The measurements included NR-$PM_1$ species by the ACSM, particle number concentrations (15 – 600 nm) by a scanning mobility particle sizer (SMPS) that is equipped with a long differential mobility analyzer (DMA 3082L, TSI) and a condensation particle counter (CPC

3775, TSI) , BC by a seven-wavelength Aethalometer (AE33, Magee Scientific Corp.) and scattering and extinction coefficients of fine particles at 630 nm by a CAPS single-scattering albedo monitor (CAPS $PM_{ssa}$, Aerodyne Research Inc.). The sampling setup and operations at 260 m were overall similar to those at ground site.

**2.3 Data analysis**

The high resolution mass spectra of HR-AMS were analyzed using the standard ToF-AMS data analysis software packages

(SQUIRREL       v.       1.57H       and       PIKA       v.       1.16H, http://cires1.colorado.edu/jimenez-group/ToFAMSResources/ToFSoftware/index.html) and the ACSM data were analyzed for the mass concentrations using the standard data analysis software (v. 1.5.3.0). The default relative ionization efficiency (RIE) values were used for $NO_3$ , $SO_4$, Chl, and Org except $NH_4$ that was determined from pure $NH_4NO_3$ particles during IE calibration.

CE was introduced to correct the incomplete detection of submicron particles by the HR-AMS and ACSM (Zhang et al., 2005),which depends on several particle and ambient air properties, including particle acidity, ammonium nitrate fraction



and relative humidity (Matthew et al., 2008;Middlebrook et al., 2012). In this study, aerosol particles were overall neutralized at both ground level and 260 m (Fig. S2). The average mass fraction of $NH_4NO_3$ was 19.4% and 18.3% at 260 m and ground site, respectively indicating that $NH_4NO_3$ would not affect CE substantially. In addition, aerosol particles were dried before sampling into the inlet system. Thus, CE=0.5 was used for both the HR-AMS and ACSM data analysis. Further

validations of CE by comparing with collocated measurements are given in Sect. 3.1.

The elemental composition of OA, including oxygen-to-carbon (O/C), hydrogen-to-carbon (H/C), nitrogen-to-carbon (N/C) and organic mass–to–organic carbon (OM/OC) ratio was determined with the "Improved-Ambient" (I-A) method (Canagaratna et al., 2015), which was revised recently on the basis of previous "Aiken-Ambient" (A-A) method (Aiken et al., 2007;Aiken et al., 2008). Note that the O/C and H/C ratios derived from the I-A method are on average 27% and 9%,

respectively, larger than those from the A-A method (Fig. S3).

Positive Matrix Factorization (PMF) with PMF2.exe (v4.2) algorithm (Paatero and Tapper, 1994) was used to analyze the HR-AMS and ACSM organic spectral matrices following the procedures reported in Ulbrich et al. (2009). The detailed data pretreatment and PMF evaluations were the same as those reported in Chen et al. (2015) and Xu et al. (2015). Two factors including a hydrocarbon-like OA (HOA) and an oxygenated OA (OOA) were identified at 260 m, while four factors

including a HOA, a cooking-related OA (COA), a more oxidized OOA (MO-OOA) and a less oxidized OOA (LO-OOA) were resolved at ground site due to the improved sensitivity and chemical resolution of the HR-AMS. These OA factors were distinguished by their unique signatures in mass spectral patterns and diurnal variations. The correlations between OA factors and specific external tracers and also other diagnostic plots were given in supplementary (Fig. S4–S7).

### 2.4 Estimation of particle density

The number concentrations from SMPS measurements were converted to mass concentrations using the composition-dependent density (Fig. 1a) that was estimated with the measured submicron aerosol composition assuming spherical particles in Eq. (1) (Aiken et al., 2009;Salcedo et al., 2006):

$$\rho_{comp} = \frac{[NO_3^-]+[SO_4^{2-}]+[NH_4^+]+[Cl^-]+[BC]+[Org]}{\frac{[NO_3^-]+[SO_4^{2-}]+[NH_4^+]}{1.75}+\frac{[Cl^-]}{1.52}+\frac{[BC]}{1.77}+\frac{[Org]}{1.2}} \tag{1}$$

where the densities are 1.75 g cm$^{-3}$ for ammonium nitrate and ammonium sulfate, 1.52 g cm$^{-3}$ for ammonium chloride (Lide,

1991), 1.2 g cm$^{-3}$ for organic aerosols (Turpin and Lim, 2001), and 1.77 g cm$^{-3}$ for BC (Park et al., 2004;Poulain et al., 2014).

### 2.5 Source region analysis

The footprints of aerosol particles during specific events were calculated using the Lagrangian particle dispersion model FLEXPART (Stohl et al., 2005). In this study, the meteorological filed was simulated by the Weather Research and Forecasting (WRF) model (Skamarock et al., 2005) with a time resolution of 1 h and spatial resolution of 10 km. In the





simulations, 2-day backward and 10000 tracer particles were calculated at a height of 50 m during five polluted episodes and two clean periods to investigate the potential source regions of aerosol particles. In addition, the potential source regions of $PM_1$ species during control and non-control periods at 260 m were investigated using bivariate polar plots (Carslaw and Ropkins, 2012). Compared with FLEXPART analysis, bivariate polar plots depending on wind speed and wind direction are

subject to offer more insights into the sources of aerosol species over a small scale.

### 3 Results and discussion

#### 3.1 Inter-comparisons

As indicated in Fig. 1, $PM_1$ (= NR-$PM_1$ + BC) overall tracked well with collocated measurements from the SMPS, TEOM, and CAPS with $R^2$ ranging from 0.84 to 0.94. The mass concentrations derived from the SMPS measurements on average

accounted for 45% of $PM_1$, mainly due to the different size cutoff (e.g., 10 – 400 nm for SMPS). On average, $PM_1$ contributed 81% of $PM_{2.5}$, which indicates that submicron aerosol accounted for a large fraction of $PM_{2.5}$ in Beijing. Note that this value is slightly higher than that obtained during previous studies conducted in urban Beijing (64%−77%) (Sun et al., 2012;Sun et al., 2015b;Zhang et al., 2014). One explanation is that BC was included into the comparisons. In addition, the light extinction of fine particles was tightly correlated with $PM_1$ at ground site ($R^2$ =0.92) as well, and the mass extinction

coefficient of $PM_{2.5}$ was also consist with previous observations in Beijing (Han et al., 2015).

The collocated measurements at 260 m were also correlated well with each other ($R^2$ = 0.76 – 0.94; Fig. S8). The $PM_1$ measured by the ACSM and AE33 showed an excellent agreement with that (15 – 600 nm) derived from the SMPS measurements ($R^2$ = 0.92, slope = 0.91; Fig. S9c). We also compared the $PM_1$ concentrations that were converted from 15 – 400 nm particles at both of these two heights (Figs. S9a and S9b). A higher ratio of 0.69 was observed at 260 m than that at

ground (0.45). This difference might suggest higher contributions of aerosol particles with aerodynamic diameters below 400 nm at 260 m than ground site. The detailed discussions of the differences in size distributions and particle number concentrations between 260 m and ground site will be presented elsewhere (Du et al., in preparation).

#### 3.2 Mass concentrations and chemical composition

The time series of $PM_1$ and meteorological parameters at two different heights during the entire study is shown in Fig. 2. The

$PM_1$ concentrations remained consistently low, generally less than 50 µg m$^{-3}$ during the control period, while episodes with $PM_1$ concentrations higher than 50 µg m$^{-3}$ were frequently observed after the control period. On average, the mass concentrations of $PM_1$ are 19.3 µg m$^{-3}$ and 14.8 µg m$^{-3}$ at ground level and 260 m, respectively during the control period, which was decreased by 57% and 50% compared to those after the control period (Table 1).

The average mass concentration of $PM_1$ at ground level during the control period was significantly lower than that during





other periods with strict emission controls, for example, 63.1 μg m$^{-3}$ during Beijing 2008 Olympics Games (Huang et al., 2010) and 41.6 μg m$^{-3}$ during 2014 APEC summit (Xu et al., 2015). These results indicate a significantly improved air quality during the parade period. Inorganic species (e.g., sulfate, nitrate and chloride) during the control period were 63%-87% lower than those measured after the control period, while organics had a relatively smaller decrease by 42%. The decreases

of aerosol species during the parade were substantially larger than those during APEC (e.g., 41-69% for inorganic species and 35% for organics) (Xu et al., 2015), likely due to more strict emission controls and the favorable northerly winds (Fig. S1a). Although the average PM$_1$ concentration (11.3 μg m$^{-3}$) measured at Tsinghua University during the same period is relatively lower than that in this study, the reductions in total PM$_1$ (63.5%), inorganic species ( 65-78%) and organics (53%) were consistent (Li et al., 2016).

Compared to ground level, the average PM$_1$ concentration at 260 m was 14.8 μg m$^{-3}$ during the control period, nearly half of that after the control period (29.8 μg m$^{-3}$). Consistently, the concentration was lower than that measured at 260 m during the 2014 APEC summit (24.1 μg m$^{-3}$) (Chen et al., 2015). The PM$_1$ species were also decreased substantially during the control period, which are 52- 69% for inorganic species and 35% for OA. Note that aerosol reductions at ground level were overall larger than those at 260 m (Table 1), indicating that local and regional emission controls might affect ground level

and 260 m differently.

The average composition of PM$_1$ at ground level and 260 m during and after the control period is shown in Fig. 2. Aerosol composition was overall similar at the two heights during the control period, which was both dominated by organics (53-55%) followed by sulfate (15-18%) and nitrate (12-15%). After the control period, aerosol composition had significant changes, particularly for OA and nitrate. For instance, the OA contribution was decreased from 53-55% to 40% while the

nitrate contribution increased from 12 – 15% to 20 – 22%. Such compositional changes were overall similar to those observed during APEC. One explanation is that regional emission controls have a large impact on the reduction of secondary aerosols by decreasing their precursors and suppressing secondary formation, while strong local OA sources, e.g., cooking activities, still exist during the control period (Xu et al., 2015). We also observed higher nitrate contributions at 260 m and higher sulfate contributions at ground level. Indeed, the average ratio of NO$_3$/SO$_4$ at ground (0.92) was lower than that

obtained at 260 m (1.29). Similar vertical differences were reported in previous studies in Beijing, which were likely caused by the favorable gas-particle partitioning of nitrate with low $T$ at 260 m (Sun et al., 2015a;Chen et al., 2015). These results also demonstrate that different sources and formation mechanisms can affect the vertical distributions of nitrate and sulfate (Sun et al., 2015a).

### 3.3 OA composition and sources

Four OA factors including two primary factors (HOA and COA) and two secondary factors (LO-OOA and MO-OOA) were identified at ground site. The HOA mass spectrum was primarily composed of hydrocarbon ions of C$_n$H$_{2n-1}^+$ and C$_n$H$_{2n+1}^+$



(Fig. 3a) that are common characteristics of primary combustion emissions, e.g., diesel and gasoline exhaust (Canagaratna et al., 2004;Lanz et al., 2007;Mohr et al., 2009). The O/C ratio of HOA (= 0.23, I-A method) was higher than the values (0.15-0.17, A-A method) reported in previous studies in Beijing (Huang et al., 2010;Xu et al., 2015) and Shanghai (Huang et al., 2012). HOA was well correlated with BC ($R^2$=0.71) during the entire study. Note that HOA was correlated with BC

during both control period ($R^2 = 0.69$) and non-control period ($R^2 = 0.58$), and the ratio of HOA/BC was also similar (0.74 vs. 0.69). These results indicate that HOA has similar sources as BC at the ground site, and the emission controls did not affect HOA/BC ratio significantly. Indeed, the concentrations of HOA and BC decreased similarly by ~57% during the control period.

The COA mass spectrum is characterized by high $m/z$ 55/57 ratio, and also has a similar O/C ratio (=0.13, I-A method) to

those (0.08 – 0.13, A-A method) from traditional Chinese cooking emissions (He et al., 2010). COA was correlated with the marker ion $C_6H_{10}O^+$ ($R^2 = 0.95$), and showed a pronounced diurnal cycle with two prominent peaks occurring at lunch and dinner times, consistent with the observations from many previous studies (Huang et al., 2010;Sun et al., 2011;Xu et al., 2014). The average COA concentration was 2.89 µg m$^{-3}$ during the control period, which was slightly lower than that after the control period (4.26 µg m$^{-3}$). In fact, Fig. 3b shows that the variations and concentrations of COA were similar between

control and non-control period. These results indicate the presence of local cooking emissions despite regional emission controls. Considering that COA contributed a comparable fraction of OA, which is 27% and 23% during control and non-control period, respectively, a stricter control of local cooking emissions is needed to further improve air quality in urban Beijing.

The two OOA factors were both dominated by prominent $m/z$ 44 yet with different O/C ratios. The O/C ratio of MO-OOA

is 1.0, which is much higher than that of LO-OOA (0.84). In addition, MO-OOA showed much higher $f_{44}$ (mass fraction of $m/z$ 44 in OA) and $m/z$ 43/44 ratio than LO-OOA. All these results indicate that MO-OOA was more oxidized than LO-OOA. Consistently, MO-OOA was better correlated with sulfate compared with LO-OOA ($R^2$: 0.86 vs. 0.64) during this study. The fact that LO-OOA was weakly correlated with nitrate during control and non-control periods suggested very different formation mechanisms between these two species. The temporal variations of MO-OOA and LO-OOA were substantially

different. As shown in Fig. 3, MO-OOA showed ubiquitously lower concentration during the control period than non-control period. On average, the MO-OOA concentration is 1.36 µg m$^{-3}$ during the control period, which is decreased by 75% compared to that after the control period. In contrast, LO-OOA showed comparable concentrations during and after the control period, with the average concentration being 5.57 and 6.48 µg m$^{-3}$. Our results showed that SOA was dominated by less oxidized OA during the control period while highly oxidized OA was more important after the control period. One

explanation is that regional emission controls slowed down the aging processes of OA by decreasing its precursors of volatile organic compounds (VOCs).




Compared to the ground site, two OA factors, i.e. HOA and OOA were identified at 260 m (Fig. 4), mainly due to low mass resolution and sensitivity of the ACSM. The mass spectral profiles of these two OA factors were similar to those identified at 260 m before and during APEC (Sun et al., 2015a;Chen et al., 2015). High *m/z* 55/57 and noticeable *m/z* 60, a marker for biomass burning in the spectrum likely indicate that HOA is a factor mixed with multiple primary emissions, e.g.,

traffic, biomass burning and cooking. HOA was tightly correlated with BC during non-control period ($R^2 = 0.70$), whereas the correlation was much weaker during the control period ($R^2 = 0.30$). Similarly stronger correlations during the non-control period were also observed between OOA and secondary inorganic species (Fig. 4). Such differences in correlations likely indicate that regional emission controls have changed the variations in BC, OA and inorganic species at higher altitudes while those at ground sites were less affected.

Overall, the composition of OA was dominated by SOA at both ground site and 260 m during the entire study, on average accounting for 65% (Fig. 5). The average mass concentrations of SOA and POA are 7.10 μg m$^{-3}$ and 3.79 μg m$^{-3}$, respectively at 260 m, which was 32.5% and 31.6% lower than those at ground site. Although the temporal variations of SOA were similar ($R^2$=0.64 and 0.77 during and after the control period, respectively) between ground level and 260 m, POA showed more dramatic differences between these two heights ($R^2$= 0.17 and 0.15 during and after the control period, respectively).

One explanation is that SOA was more influenced by the formation over regional scales, while the vertical differences in POA were more sensitive to various local primary source emissions, including cooking, traffic and biomass burning. The fact that the diurnal variations of POA varied more dramatically at ground site than 260 m supported that ground site is more subject to influences from local sources compared with 260 m (Chen et al., 2015). Indeed, POA was even correlated with SOA at 260 m ($R^2$=0.69) and the diurnal OA composition was relatively stable throughout the day, indicating that primary

and secondary species were more well mixed at higher altitudes due to a dominant source from regional transport. Although the contributions of POA and SOA to OA remained small changes during and after control periods at ground site and 260 m, SOA composition at ground site had significant changes. While the contribution of LO-OOA to OA was decreased from 52% during the control period to 35% after the control period, that of MO-OOA was increased from 13% to 30%. These changes in SOA composition illustrated the importance of fresh SOA during clean periods, while aged SOA is more important during

polluted episodes. Further studies are needed to investigate how emission controls and meteorological differences affect the transformations between LO-OOA and MO-OOA.

### 3.4 Aerosol chemistry differences between ground level and 260 m

As indicated in Fig. 2c, the ratio of 260 m to ground ($R_{260m/Ground}$) for PM$_1$ was less than 1 for most of the time, which indicated that the PM$_1$ level at 260 m was overall lower than that at ground level. Indeed, NR-PM$_1$ species measured at 260

m were 35-53% lower than those at ground level (Fig. S10). Although PM$_1$ species showed similar variations between the two heights ($R^2$ = 0.59-0.80), periods with largely different concentrations were also observed, particularly during the periods





with high PM$_1$ levels (Fig. 6). Fig. 7 depicts that $R_{260m/Ground}$ showed a gradual decrease as a function of PM loadings (> 60 µg m$^{-3}$) for most aerosol species, e.g., sulfate, nitrate, and ammonium, indicating larger vertical gradients during more polluted episodes. For example, the $R_{260m/Ground}$ of nitrate was approximately 0.88 below 90 µg m$^{-3}$, and then rapidly decreased to 0.28 when PM$_1$ increased to 210 µg m$^{-3}$. Also note that the $R_{260m/Ground}$ varied substantially different for each

species. Sulfate showed the lowest $R_{260m/Ground}$ among all aerosol species during the same PM levels. One explanation is the different sources at different heights. Another explanation is the different formation mechanisms associated with different meteorological parameters, e.g., RH, $T$, and solar radiation, and also the difference size distributions at the two heights. POA showed a largely different vertical variation from SOA. Previous studies showed overall higher POA concentrations at ground level due to the influences from local source emissions (Sun et al., 2015a). However, POA at 260 m was comparable

with that at ground level in this study. One reason is that POA at 260 m was mixed with part of SOA due to the limitation of PMF analysis of ACSM spectra. This is supported by the similar variations between POA and SOA at 260 m ($R^2$ =0.69). BC is a clear exception showing an increasing trend in $R_{260m/Ground}$ as a function of PM loadings. The reason is not clear yet, but could be due to more aged BC at 260 m from regional transport while local BC emissions are not significant. For example, diesel trucks and heavy duty-vehicles are not allowed inside the city during daytime. Figure 7 also showed a large variation

in $R_{260m/Ground}$ during periods with low PM loadings (e.g., < 30 µg m$^{-3}$ ) as indicated by the range of 25$^{th}$ and 75$^{th}$ percentiles. This result indicates that the vertical differences between ground level and 260 m during clean periods are subject to multiple influences and can vary substantially. In addition, the evolution of mixing layer could be another factor that influences $R_{260m/Ground}$. As shown in Fig. S11, almost all aerosol species showed clear diurnal patterns with lower $R_{260m/Ground}$ values at nighttime and in the early morning and higher values during afternoon. Sulfate however showed a much smaller variation in

diurnal cycle of $R_{260m/Ground}$, consistent with the fact that sulfate was formed over a regional scale and relatively well mixed.

**3.5 Diurnal evolution of aerosol species**

The diurnal variations of PM$_1$ species during and after the control periods at ground level and 260 m are depicted in Fig. 8. While the diurnal patterns between the two heights were overall similar, we also observed large vertical differences, particularly at nighttime. For example, POA in OA showed much larger vertical differences at nighttime than daytime due to

enhanced cooking emissions and shallow boundary layer. As a comparison, such differences were much reduced during daytime due to a stronger vertical mixing associated with elevated boundary layer height.

SOA showed similar diurnal variations between the two heights, yet the diurnal variations were quite different during and after the control period. SOA showed a pronounced noon peak during the control period indicating the importance of photochemical processing in driving the diurnal evolution of SOA. This also explained the dominance of photo-chemically

processed LO-OOA in SOA during the control period. In contrast, the diurnal cycle of SOA were relatively flat after the control period, likely indicating that SOA was more aged over a regional scale, which is consistent with the dominance of



highly oxidized MO-OOA in SOA. The diurnal variations of POA and SOA together led to a strong diurnal cycle of organics at ground level and 260 m, which was characterized by two pronounced peaks at noon time and nighttime. The diurnal cycle of sulfate was remarkably similar between ground level and 260 m. This is consistent with the fact that sulfate is mainly formed over a regional scale, and the contribution of local photochemical production is generally small (Salcedo et al.,

2006;Sun et al., 2011). The diurnal cycle of BC, a tracer of combustion emissions, varied differently from secondary aerosol species, but was generally characterized by high concentrations at nighttime (Han et al., 2009;Huang et al., 2010). As shown in Fig. 8h, BC showed higher concentrations at ground site than 260 m at nighttime during the control period, while it was reversed during daytime. One explanation is that BC was dominantly from local diesel trucks and heavy-duty vehicles emissions at nighttime, which can cause a clear vertical gradient due to the weaker vertical convection. In contrast, BC at

260 m was generally higher than that at ground site during non-control period. This likely indicates an important source of BC being from regional transport (Wang et al., 2016).

Ammonium, nitrate and chloride showed similar diurnal patterns at ground level and 260 m. However, the diurnal patterns were substantially different during and after the control period, indicating the changes in either sources or formation mechanisms. During the control period, the three species showed much higher concentrations at nighttime than daytime,

suggesting a dominant impact of temperature dependent gas-particle partitioning on diurnal variations. Note that chloride presented higher concentrations at 260 m than ground site during the control period. One reason is likely due to the uncertainties in ACSM measurements because the concentrations were close to 5 min detection limits (0.07 μg m$^{-3}$ )(Sun et al., 2012). After the control period, these three species presented the highest concentrations from midnight to 10:00 at both ground site and 260 m, and then rapidly decreased to low levels during the late afternoon. Such diurnal behaviors were

similar to those previously observed in Beijing (Huang et al., 2010;Sun et al., 2012), which were mainly driven by temperature dependent gas-particle partitioning. Vertically, the differences between ground site and 260 m were the largest from midnight to early morning, while they were almost disappeared in the afternoon due to strong vertical mixing.

### 3.6 Potential sources of aerosol species

Bivariate polar plots were used to explore potential source regions of submicron aerosol species at 260 m during control and

non-control periods (Carslaw and Ropkins, 2012). As shown in Fig. 9a, high mass concentrations of aerosol species were mainly located in regions to the south and southeast of the observation site, highlighting an important role of regional transport to relatively high PM levels during the control period. Moreover, the center of bivariate polar plots with low wind speeds showed ubiquitously low concentrations, suggesting the absence of stagnant meteorological conditions in accumulation of pollutants during the control period. We also noticed different source regions for different aerosol species.

For example, the source regions of BC, chloride, nitrate and ammonium were mainly located to the southeast, while POA, SOA and sulfate were from both the south and southeast. These results also indicate that regional emission controls likely





changed aerosol composition substantially to the south and southeast Beijing.

The potential source regions of aerosol species after the control period (Fig. 9b) were quite different from those during the control period. Most aerosol species showed high concentrations in the regions to the south, indicating the importance of regional transport in the formation of haze episodes. BC, chloride and POA however also showed high concentrations in

areas with low wind speed, likely suggesting a considerable contribution of local sources emissions under stagnant meteorological conditions. Our results have significant implications that air pollution mitigating strategies should focus on emission controls in regions to the south and southeast of Beijing while strict local emission controls are also needed to reduce primary species.

### 3.7 Case studies to investigate the effects of emission controls

Five polluted episodes and two clean periods (Fig. 6) were selected to better investigate the impacts of emission controls on aerosol chemistry during the control period. The two clean periods C1 and C2 showed similar footprint regions (Fig. 10) that were dominantly from the northeast. However, the average mass concentrations of $PM_1$ during the control period (C1) were even higher by 33% and 22% at ground level and 260 m, respectively than those after the control period (C2, Table S1). We note that RH during C1 was much higher than that during C2 due to the influence of a precipitation event before C1.

Therefore, higher mass concentrations during C1 were likely due to higher RH that facilitated the formation of secondary aerosols. In fact, the mass concentrations of SIA during C1 were $1.4 - 1.5$ and $1.6 - 2.1$ times at 260 m and ground level, respectively of those during C2, whereas primary species, e.g., BC and POA were relatively close between C1 and C2 by varying less than 10%. Our results indicate that regional emission controls during parade appeared to have small impacts on clean periods for two reasons: (1) the air masses during clean periods are generally from the relatively clean regions with low

anthropogenic emissions in the northwest and northeast, and (2) regional emission controls were mainly implemented in the regions to the south and southeast, e.g., Hebei, Tianjin and Shandong provinces.

The footprint regions varied differently among the five polluted episodes, but generally from the south and southeast except PE1 (Fig. 10). The two episodes, PE1 and PE2 during the control period showed substantially different composition which was likely due to the different source regions given their relatively similar meteorological conditions (Table S1). The

average mass concentration of $PM_1$ during PE1 was ~40% higher than that during PE2 at both ground level and 260 m. While the $PM_1$ during PE1 was mainly composed of SIA (65% and 59% at ground level and 260 m, respectively), it was characterized by higher contribution of organics (44% and 47% at ground level and 260 m, respectively) during PE2. These results suggest that aerosol composition and concentrations during the control period can have significant differences during polluted episodes from different source regions. The mass concentrations of $PM_1$ varied more dramatically during three

polluted episodes after the control period. While the average $PM_1$ mass concentration was close between PE3 and PE4, it was significantly lower than that during PE5. By comparing with PE1 and PE2 during the control period, we found that PE4



showed the most similar footprint region as PE2, and also the meteorological conditions between PE4 and PE2 were relatively similar. Therefore, the comparisons between PE2 and PE4 would be the best case to illustrate the impacts of emission controls in the south on PM levels and aerosol composition during parade. The average $PM_1$ concentration was 30.1 and 25.2 $\mu g\ m^{-3}$ at ground level and 260 m, respectively during PE2, which was reduced by 44-45% compared with those during PE4. Furthermore, we observed remarkably similar composition of $PM_1$ and OA at 260 m between PE2 and PE4, which comprised 46% SIA and 66% SOA, respectively. This finding is consistent with our previous studies that regional emission controls can reduce PM levels substantially but have small impacts in changing the bulk composition due to the synergetic controls of aerosol precursors (Chen et al., 2015). In contrast, aerosol composition was slightly different between PE2 and PE4 at ground level. Primary species, e.g., BC, Chl, and POA showed relatively more reductions (55 – 67%) than secondary aerosol species (33% - 44%), indicating that local emission controls also played a role in reducing the PM levels at ground sites.

**4 Conclusions**

A HR-AMS and an ACSM were deployed at ground level and 260 m on a meteorological tower, respectively for real-time characterization of submicron aerosol chemistry in Beijing from 22 August to 30 September 2015. Our results showed that regional emission controls were effective in reducing $PM_1$ levels by 57% and 50% at ground level and 260 m, respectively during the control period. Aerosol composition was also substaintially different during and after the control period. While OA contributed the major fraction of $PM_1$ at both ground site and 260 m (55% and 53%, respectively) during the control period, its contribution was decreased to 40% associated with an increase in SIA. These results illustrated a larger impact of emissions controls on SIA reduction than OA, consistent with our previous finds during the APEC. The responses of POA and SOA to emission controls were investigated by positive matrix factorization analysis of OA. We found that the total POA decreased similarly as the total SOA at ground level (40% vs. 42%) and 260 m (35% vs. 36%) during the control period. However, the SOA composition at ground level had substantial changes during and after the control period. While the contribution of LO-OOA decreased from 52% to 35%, MO-OOA increased from 13% to 30%. It appears that photochemal produciton of less oxidied SOA was the domiant SOA formation mechanism during the control period, while aged SOA was more signficant after the control period. Our results highlighted that regional emission controls not only reduces PM levels substaintially, but likely affect SOA formation mechanisms as well. By comparing two polluted episodes with similar footprint regions during and after the control period, we found that regional emission controls reduced $PM_1$ levels by 44-45% at different heights, and primary species had relatively more reductions (55 – 67%) than secondary species (33% - 44%) at ground level. The vertical differences for aerosol species between ground level and 260 m during and after the control period were also investigated. We found that $R_{260m/Ground}$ showed gradual decreases as a function of PM levels for most aerosol





species during high loading periods ($> 60$ µg m$^{-3}$), indicating larger vertical gradients during more polluted episodes. In addition, $R_{260m/Ground}$ varied differently for primary and secondary species elucidating the different impacts of local emissions and regional transport on aerosol chemistry at different heights in the city.

*Acknowledgements.* This work was supported by the National Key Project of Basic Research (2014CB447900, 2013CB955801), the National Natural Science Foundation of China (41575120), the Special Fund for Environmental Protection Research in the Public Interest (201409001), and the Strategic Priority Research Program (B) of the Chinese Academy of Sciences (XDB05020501).

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



**Table 1.** Summary of average meteorological parameters and chemical compositions during and after the control period, and the entire study at ground level and 260 m. Change percentages in mass concentrations of PM$_1$ species during and after the control period are also shown.

| | 260 m | | | | | Ground | | | |
| --- | --- | --- | --- | --- | --- | --- | --- | --- | --- |
| | Entire | Control Period | non-control Period | Change Perc. (%) | | Entire | Control Period | non-control Period | Change Perc. (%) |
| **Meteorological Parameters** | | | | | | | | | |
| T(ºC) | 21.01 | 23.40 | 19.94 | | | 22.77 | 25.50 | 21.55 | |
| RH (%) | 66.36 | 65.73 | 66.64 | | | 62.66 | 60.18 | 63.76 | |
| WS (m s$^{-1}$) | 4.07 | 4.03 | 4.08 | | | 1.22 | 1.28 | 1.20 | |
| **Aerosol Species (µg m$^{-3}$)** | | | | | | | | | |
| Org | 10.82 | 7.77 | 12.03 | 35.4 | | 15.96 | 10.71 | 18.31 | 41.5 |
| POA | 3.77 | 2.72 | 4.18 | 34.7 | | 5.51 | 3.78 | 6.34 | 40.4 |
| SOA | 7.05 | 5.04 | 7.85 | 35.8 | | 10.45 | 6.93 | 11.97 | 42.1 |
| LO-OOA | | | | | | 6.21 | 5.57 | 6.48 | 14.0 |
| MO-OOA | | | | | | 4.24 | 1.36 | 5.49 | 75.2 |
| SO$_4$ | 3.87 | 2.19 | 4.54 | 51.8 | | 7.37 | 3.39 | 9.09 | 62.8 |
| NO$_3$ | 5.28 | 2.26 | 6.47 | 65.0 | | 7.16 | 2.37 | 9.23 | 74.3 |
| NH$_4$ | 2.74 | 1.38 | 3.29 | 58.1 | | 4.36 | 1.63 | 5.53 | 70.5 |
| Chl | 0.34 | 0.13 | 0.42 | 69.5 | | 0.37 | 0.07 | 0.51 | 87.1 |
| BC | 2.42 | 1.04 | 3.02 | 65.5 | | 2.22 | 1.17 | 2.71 | 56.6 |
| PM$_1$ | 25.47 | 14.76 | 29.76 | 50.4 | | 37.44 | 19.34 | 45.38 | 57.4 |

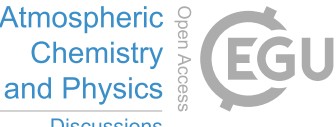


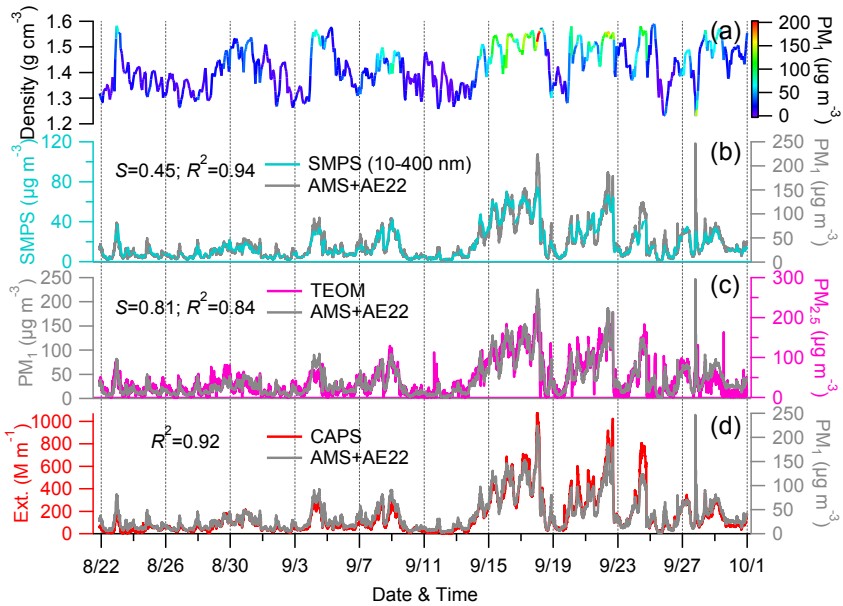

**Figure 1.** (a) Time series of particle density, and inter-comparisons between $PM_1$ (= $NR$-$PM_1$ + BC) and (b) mass concentrations derived from SMPS measurements, (c) $PM_{2.5}$ and (d) light extinction coefficient of $PM_{2.5}$. The correlation coefficients ($R^2$) and regression slopes ($S$) are also given in (b-d).

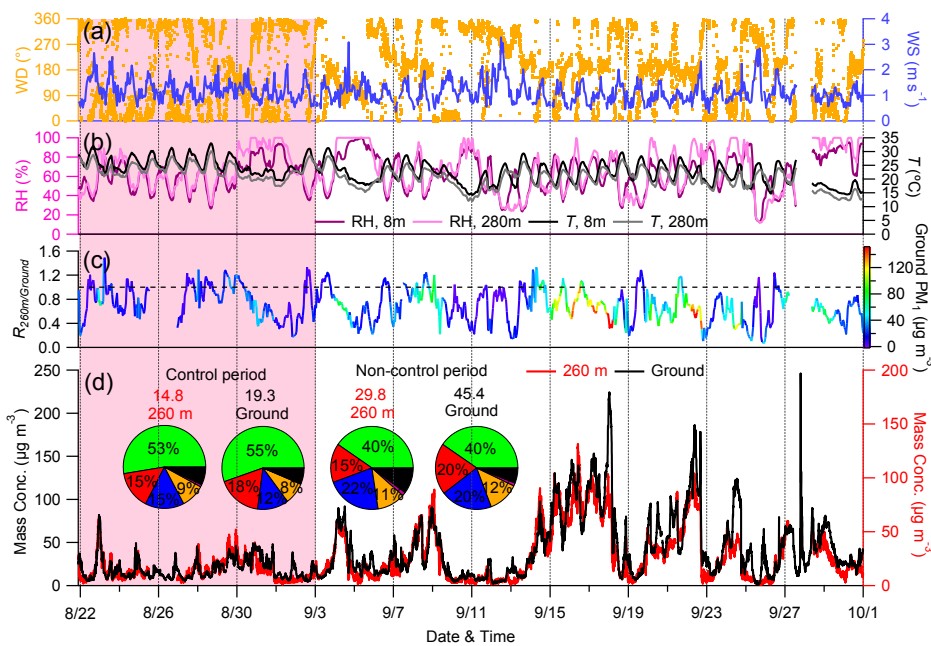

**Figure 2.** Time series of (a) wind direction (WD, 280m) and wind speed (WS, ground), (b) relative humidity (RH) and temperature ($T$), (c) ratio of 260 m to ground for $PM_1$, and (d) $PM_1$ mass concentrations at 260 m and ground. The pie charts depict the average chemical composition during and after the control period at 260 m and ground site, respectively.





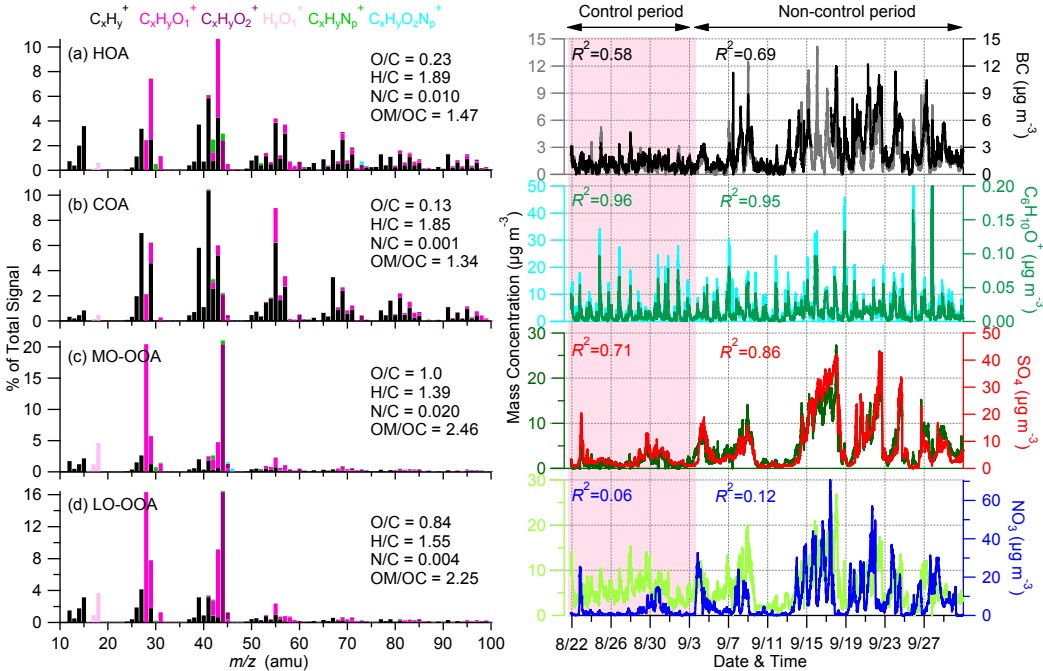

**Figure 3.** High resolution mass spectra (left panel) and time series (right panel) of four OA factors at ground site: (a) hydrocarbon-like OA (HOA), (b) cooking-related OA (COA), (c) more oxidized oxygenated OA (MO-OOA) and (d) less oxidized OOA (LO-OOA). The time series of BC, sulfate and nitrate are shown for comparisons, and the correlation coefficients ($R^2$) between OA factors and external tracers during the control (pink area) and non-control periods are also shown. The elemental ratios in the left panel were calculated from the I-A method (Canagaratna et al., 2015).



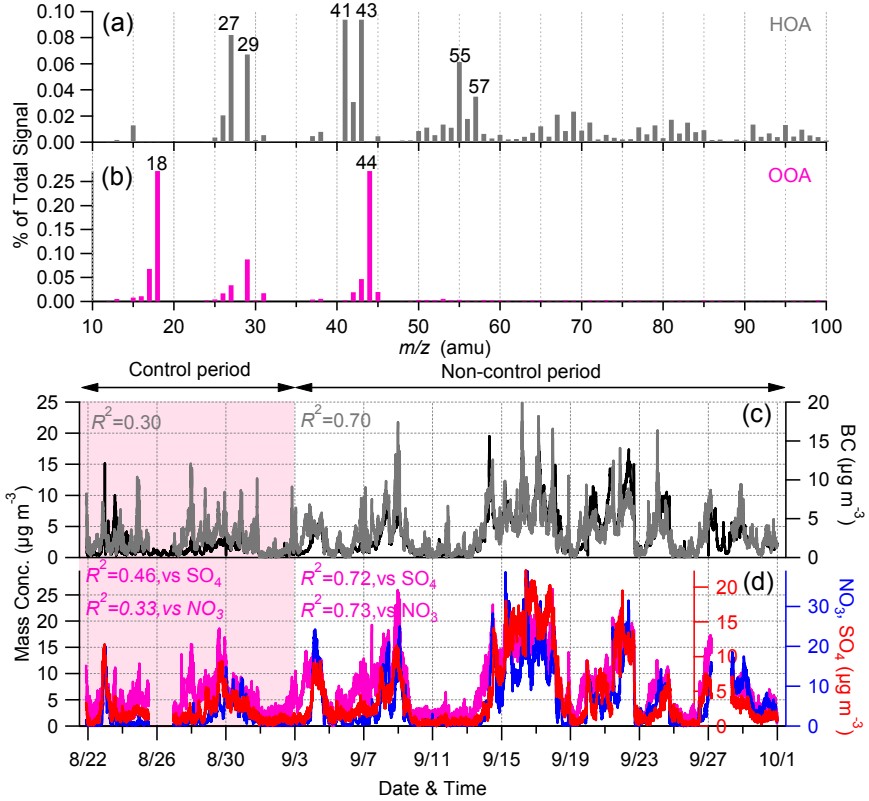

**Figure 4.** Mass spectral profiles and time series of (a, c) hydrocarbon-like OA (HOA) and (b, d) oxygenated OA (OOA) at 260 m. Also shown are the time series of BC, sulfate and nitrate for comparisons. The correlation coefficients ($R^2$) between OA factors and external tracers during control (pink area) and non-control periods are also given in panel (c) and (d).



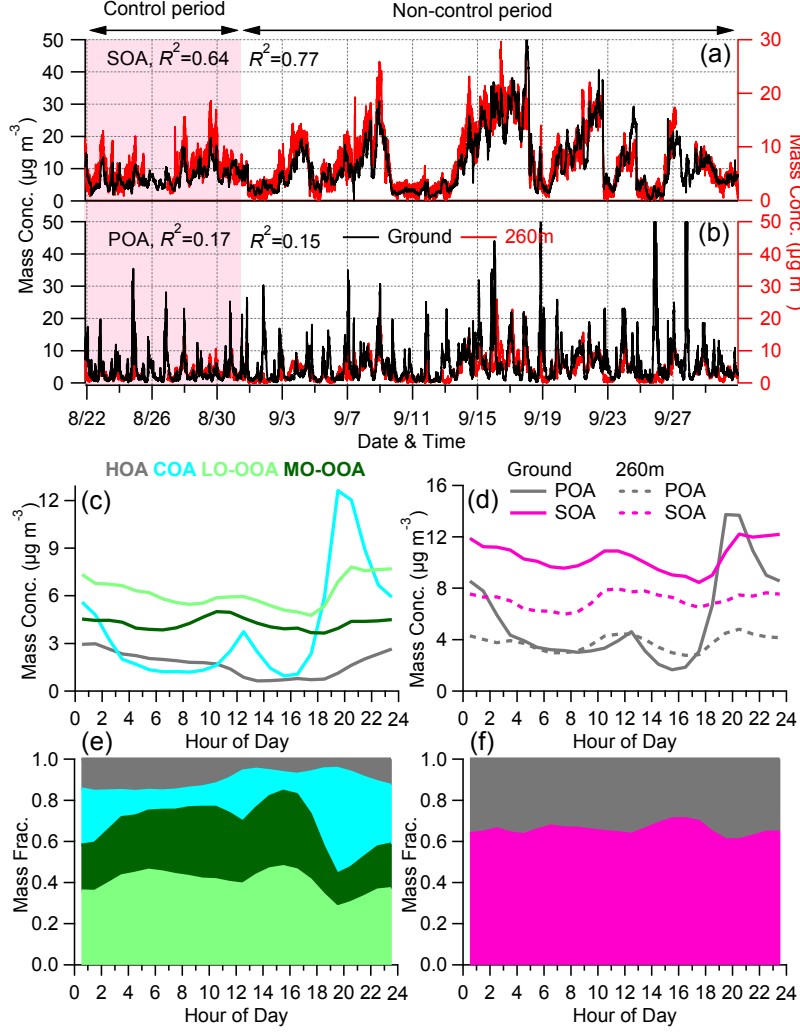

**Figure 5.** (a, b) A comparison of time series of SOA and POA at ground and 260 m, (c) diurnal variations of four OA factors at ground site, (d) diurnal variations of POA and SOA at ground site and 260 m, and (e-f) shows the diurnal mass fractions of OA factors in (c) at ground and (d) at 260 m.




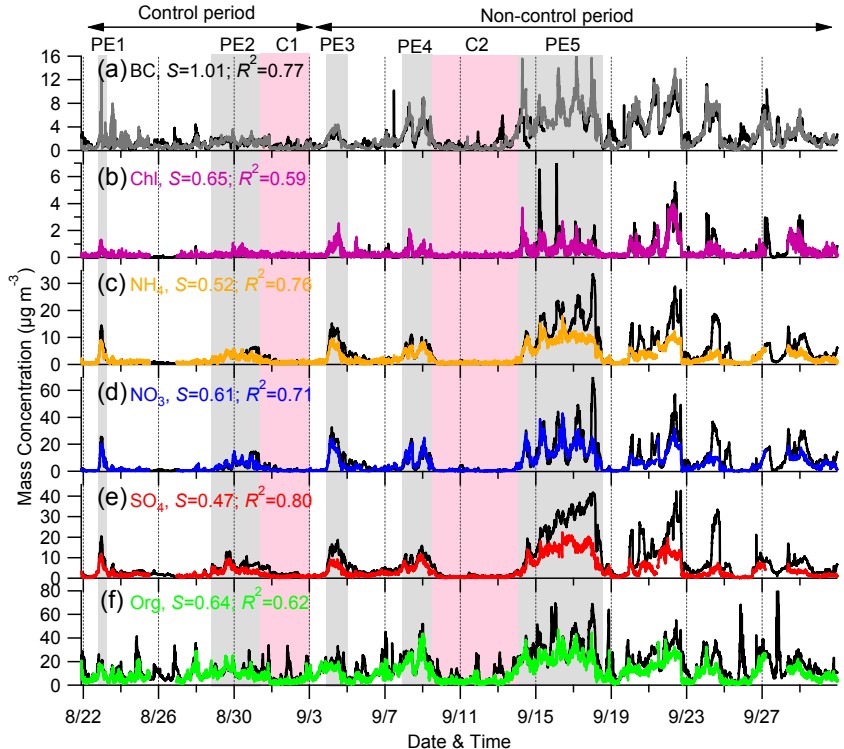

**Figure 6.** Comparisons of the time series of aerosol species between ground site (black lines) and 260 m (colored lines). Five pollution episodes (PE1, PE2, PE3, PE4 and PE5) and two clean periods (C1 and C2) are marked for further discussions.

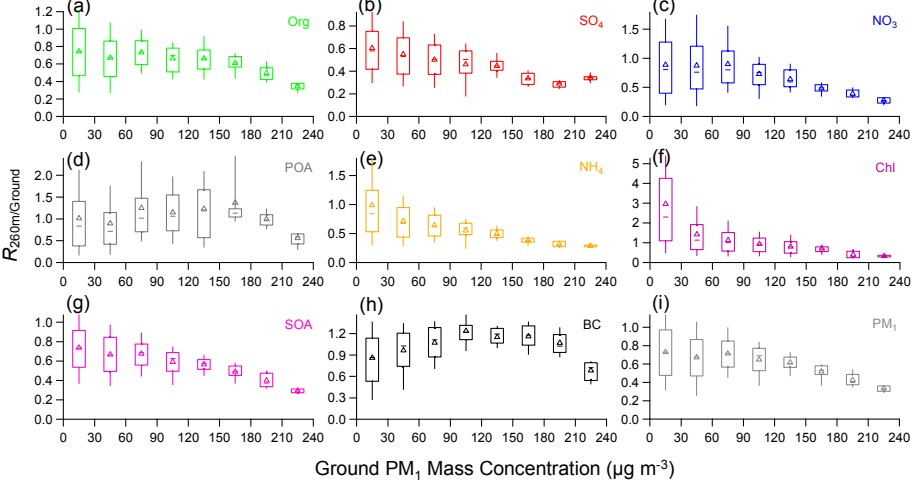

5   **Figure 7.** Variations of the ratios of 260 m to ground ($R_{260m/Ground}$) for each species as a function of $PM_1$ mass concentrations at ground level. The data are grouped in eight bins according to $PM_1$ loadings (30 µg m$^{-3}$ increment). The median (middle horizontal lines), mean (triangles), 25th and 75th percentiles (bottom and top boxes) and 10th and 90th percentiles (bottom and top whiskers) are shown for each bin.





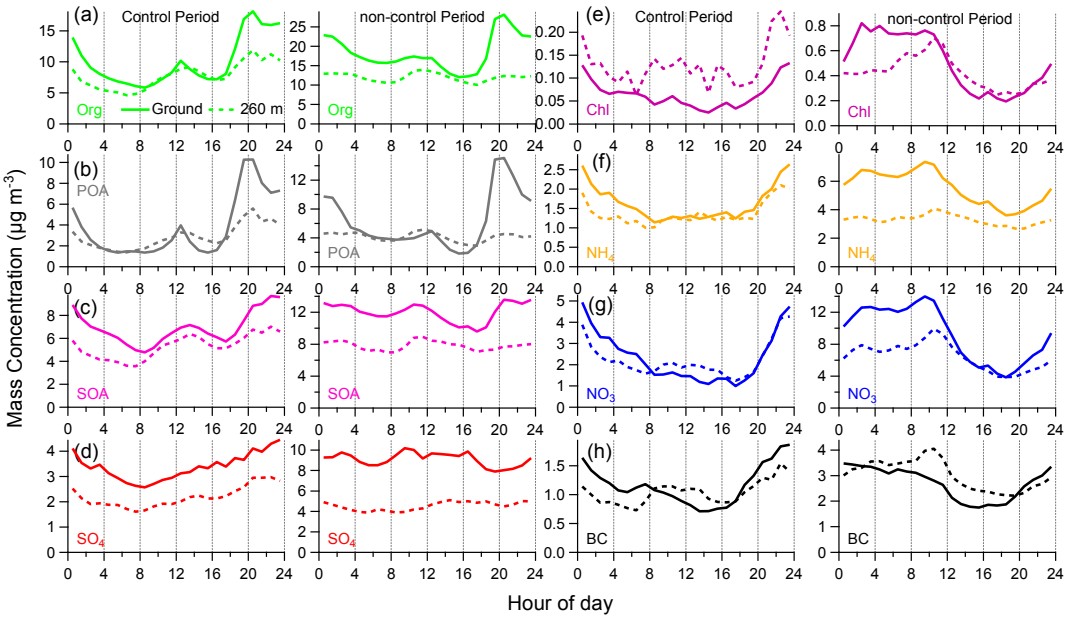

**Figure 8.** Diurnal variations of $PM_1$ species at ground level (solid lines) and 260 m (dash lines) during the control and non-control periods.





**Figure 9.** Bivariate polar plots of PM$_1$ species during the (a) control and (b) non-control periods at 260 m as functions of wind direction and wind speed (m/s).





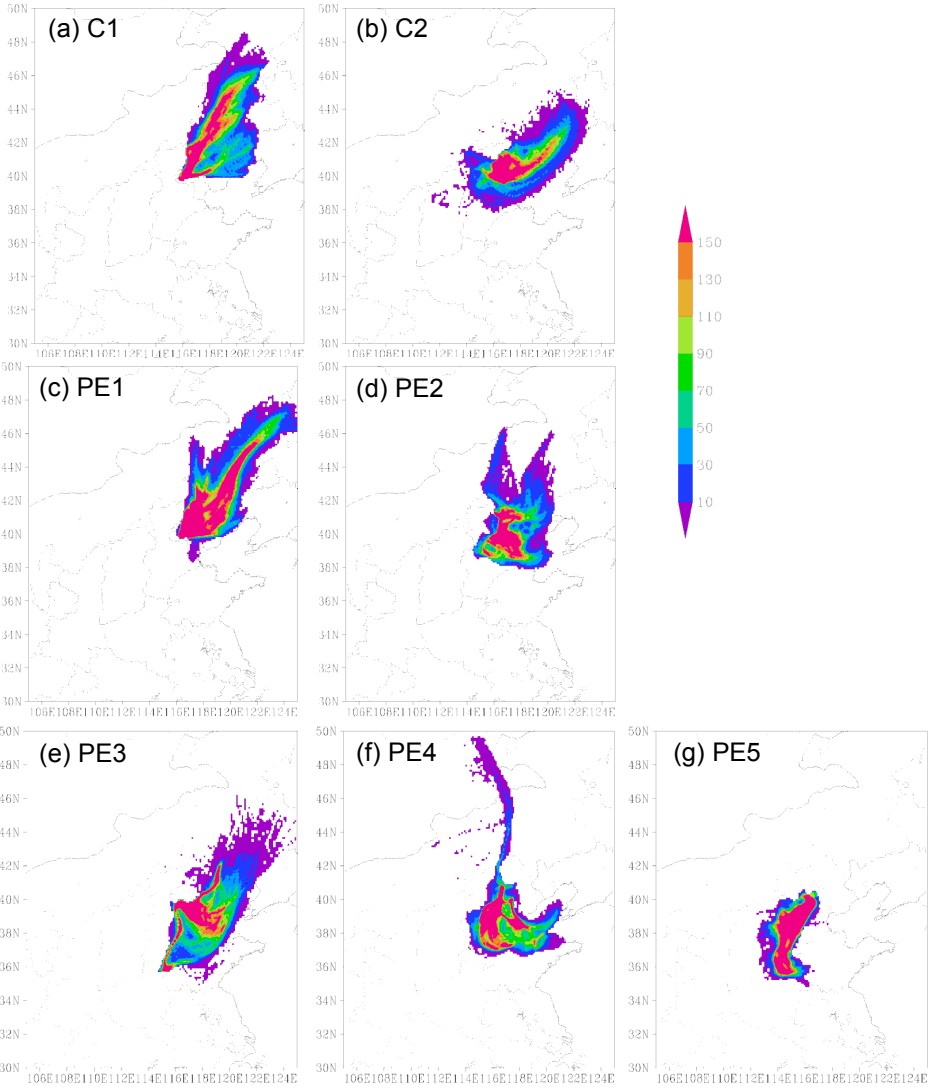

**Figure 10.** Footprint regions of two clean periods and five pollution episodes that are marked in Fig. 6. The color bar indicates the number concentrations of tracer particles.





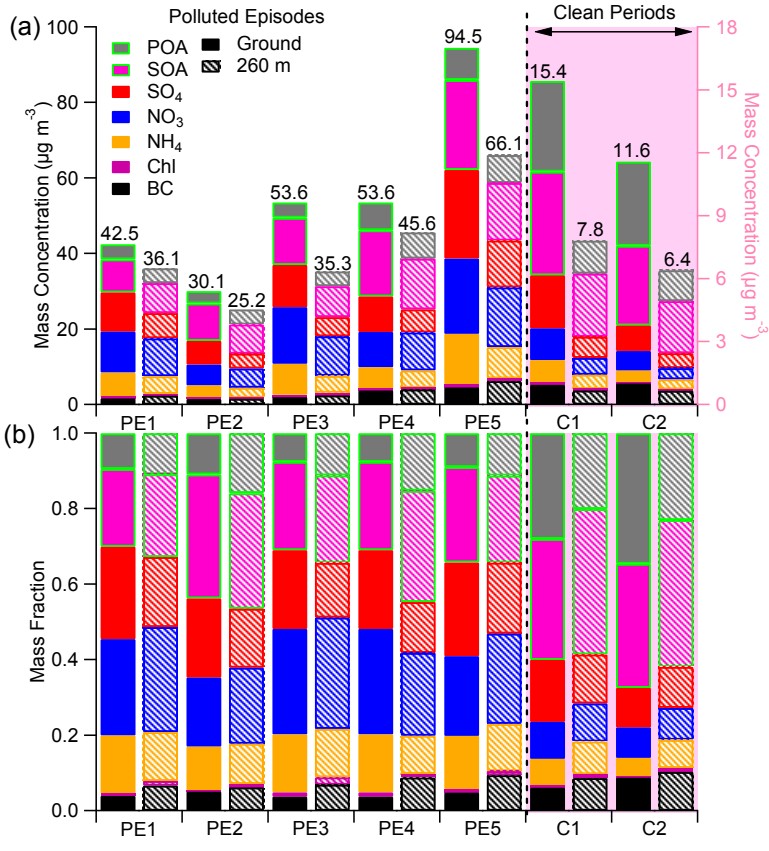

**Figure 11.** Average composition of $PM_1$: (a) mass concentrations and (b) mass fraction during the five polluted episodes (PE1, PE2, PE3, PE4 and PE5) and two clean periods (C1 and C2) at 260 m and at ground.