# Peer review of "Insights into aerosol chemistry during the 2015 China victory day parade: results from simultaneous measurements at ground level and 260 m in Beijing"

_Atmospheric Chemistry and Physics, 2016_

## Short Comment (SC1) · 21 Sep 2016

This study investigates the chemical composition and potential formation pathways of aerosols at ground level and 260 m during and after the control period in Beijing. The authors provide detailed analysis for the chemical composition and evolution data. The contribution and the effect of local control and transportation are emphasized in this study, providing precious assessment for the regional emission control impact on haze treatment. This paper should be considered for the publication in ACP if the following issues are further stated.

Major issues,

In section 3.2 and section 3.3, the authors mainly talk about the difference between mass concentration of different species during and after the control period. The authors then stated the importance of emission control on air quality. However, as the main pollution level driver, the meteorological difference between two periods are merely talked about in these two part. Then, in Page 12, Line 28-29, the author mentioned that "the absence of the stagnant meteorological conditions during the control period", the difference of the meteorological background during two periods could bias or exaggerate the effect of emission control. However, the author failed to clarify the effect of emission control when excluding the meteorological differences.

Page 9, Line 15-18, I personally disagree with the authors' statement about COA control. According to the data provided, COA was 2.89 $\mu$g m-3 during control period averagely, which is 32% lower than that during after control period. This is actually a lot decrease. It is true that COA emission has not been overall controlled and managed, the difference of concentration between these two periods should imply the difference of accumulation process of pollutants and/or meteorological conditions.

Page 9, Line 30, "regional emission controls slowed down the aging processes of OA by decreasing its precursors of volatile organic compounds"? Is there any lab/field study support for this theory?

Minor issues,

Page 2, Line 3, Huang's nature and Guo's PNAS are analysis of a relatively short scale of time of air quality compared to the authors statement "air pollution during the past decade". It is obviously inappropriate to only cite these two articles for this statement.

Page 6, Line 20-25, Why the chloride concentration is directly divided by the density of ammonium chloride?

Page 14, Line 23, misspell "photochemal"

Page 21, Figure 2, what do different colours of pie charts mean?

---

## Referee Comment (RC1) · Anonymous Referee #1 · 5 Oct 2016

The work of Zhao et al. presents PM1 measurement results during and after the 2015 China Victory parade event at ground level and 260m level using a HR-AMS and an ACSM. The paper was generally well written and results are valuable to the academia and goverment regarding air pollution control in megacities, such as Beijing. The reviewer finds a few issues, hopefully they can be well addressed before publication, as follows

(1)As the authors have published a couple of paper regarding the HR-AMS measurement at ground site, ACSM at 260 level, and one combining results at these two heights

[Figure]

during APEC event, and also another paper simply compared the measurement results at two heights in 2014, it might be better to put a bit more info to compare results in this work and these previous works, focusing on the differences rather than similiarities, to let readers know clearly the new findings of this work. (2)Section 3.1, NR-PM1 occupied ∼81% of PM2.5 mass, this value is relatively higher than ones reported in Lanzhou (cite: Atmos. Chem. Phys., 14, 12593-12611), Nanjing(cite: Atmos. Chem. Phys., 16, 9109-9127,) and previous values in Beijing. It is likely that addition of BC increase the ratio, but the reviewer feels more discussion are needed. For example, does this ratio increase or decrease with the total PM2.5 mass loading?ãĂĂProbably, at high PM2.5 loadings, the mass fractions of supermicron meter particles increased, while at relatively clean periods, more secondarily formed species reside in submicronmeter range? It might be interesting to check. (3)I believe the measurement uncertainties from HR-AMS and ACSM were constrained before their deployments, correct? it is not clear in the manuscript. For example, if the same air mass is loaded into these two instruments simutaneously, these two instruments should give the same concentrations for different species? correct? (4)Did the authors try to do PMF analyses individually on control period and non-control period?Although I understand the amount of data may be not enough in particular for the control period to conduct a robust PMF analyses, but it may be worth a try.
* * *

---

## Referee Comment (RC2) · Anonymous Referee #3 · 6 Oct 2016

This manuscript presents a comprehensive study using a suit of on-line instruments aiming to describe the air quality improvement under the emission control and the vertical distribution of particulate matter in Beijing during the 2015 China victory day parade. The results show that the mass concentration of PM1, during and after the parade, are significant different ($\sim$50% decreased) and the chemical composition and mass concentration at ground site and 260m tower general varied synchronously, suggesting the ground site also representing a regional signal. These results are very useful for validating the strategies of emission control and evaluating the radiation forcing of PM in the boundary layer in the future. The topic in this manuscript is fitted with

the range of ACP and the paper is also well written, and the results present an interest for the scientific community. This paper should be accepted on completion of the minor revisions/clarification requested below. Major comments 1. It is interesting to compare the mass variation of each species between ground and 260m site which is useful to know the respective of ground observation. As shown in Fig. 6, all species at both sites generally display similar trends. One suggestion is that add a scatter plot in each species following the time series. 2. The explanation for the comparison of BC between two heights is somewhat not convincible. The uncertainties of these two aethalometers were not presented in the measurement section. Which wavelength results were used for each aethalometers? Are the data corrected for shadow effect and accumulation effect?

Minor comments P6, L1-2: It seems that PM1 were not total neutralized based on the scatter plots between measured and predicted ammonium (slop = 0.85-0.88). P6: The formula for calculating the density of PM1 is wrong in the denominator. It should be [$NH_4NO_3$], [($NH_4)_2SO_4$], and [$NH_4Cl$], other than [$NO_3$], [$SO_4$], and [$Cl$]. P8, L7-9: Please add the information of the location of Tsinghua University and the instrument used for this study. P8, L10: The content of this sentence is somewhat duplicated with previous sentences (P7, L27-28). P8, L21-23: This explanation is too general for explaining the phenomena of increased of nitrate and decreased of OA. I suggest that you can check what kinds of sources of NOx have been closed during control period and how is the change of the level of O3. P9, L13: The variation of mass concentration of COA is more than 30% higher during after control period than control period, which is not slightly. P10, L10: Are the average contributions of SOA to OA at both ground sit and 260 m all 65%? P11, L12-13: For the vertical variation of BC, does the uncertainty of the instruments in these two heights account for the strange variation? Please add some information for these two Aethalometers measurement in the section 2.2.

---

## Referee Comment (RC3) · Anonymous Referee #2 · 7 Oct 2016

**Reviewer comment on**

**Insights into aerosol chemistry during the 2015 China victory day parade: results from simultaneous measurements at ground level and 260 m in Beijing**

*By Zhao et al., Atmos. Chem. Phys. Discuss., doi:10.5194/acp-2016-695, 2016*

**Anonymous referee #2**

**General comments:**

This manuscript reports results obtained during a field campaign undertaken at Beijing during five weeks in summer 2015. Measurements took place during and after a period of strict emission controls implemented by the Chinese Authorities to ensure good air quality in Beijing during the China victory day parade. Results clearly show an improvement of the air quality during the emission control period.

The methodology presented in this manuscript is very similar to the "APEC Blue" paper published recently by the same group (Sun et al., 2016): a high-resolution time-of-flight aerosol mass spectrometer (HR-ToF-AMS) deployed at a ground site, an aerosol chemical speciation monitor (ACSM) at 260 m on a meteorological tower, and a comparison of the particle concentration and chemical composition during and after the emission control period. However, the air quality issues faced by the inhabitants of Beijing are really impressive, so this kind of studies is of prime importance to assess the efficiency of the emission controls implemented by the Chinese Authorities. I warmly recommend the publication of this manuscript after the authors address the following comments.

**Specific comments:**

1) The authors used two different instruments measuring particle size distributions, i.e. the AMS and the SMPS. Each time that the authors give a particle diameter, I think it would be important to mention whether it corresponds to mobility diameter ($D_m$) or vacuum aerodynamic diameter ($D_{va}$).

2) Page 6, line 23: the authors used the chemical composition of the AMS to calculate the density. I'm wondering whether they also made the scatterplot of the AMS total mass vs. SMPS volume. If the size cut-off of the instruments is not the same, they can use the PToF data of the AMS, and integrate the AMS size distributions over the same size range as the SMPS.

3) Page 6, line 25: for the density of organic aerosols, the authors used the value given by Turpin and Lim (2001) (1.2 $g/cm^3$). However, they can also use the formula given by Kuwata et al. (2012) to calculate the density of organics by using elemental ratios (O/C and H/C).

4) Page 8, lines 23-26: the authors mention that the nitrate contribution to $PM_1$ was higher at 260 m than at ground level, and suggest that this result might be due to favorable gas-particle partitioning of nitrate with low temperature at 260 m. However, according to Table 1, the difference of temperature between ground level and 260 m was less than 2°C. Is this sufficient to have a significant impact on the gas-particle partitioning of nitrate?

5) Page 9, lines 19-31: there is a long discussion about the more oxidized- and less oxidized-OOA. The main problem here is that two important data are missing in this manuscript: the volatile

organic compounds (VOCs) and the solar radiation. Without this information, we don't have any idea on the reason for which the MO-OOA was significantly lower during the emission control period. This could be due to the reduction of VOCs emissions (as mentioned several times by the authors) but also to reduced photochemistry.

6) Page 12, line 15: the authors mention that the diurnal patterns observed for ammonium, nitrate, and chloride were mainly due to temperature dependent gas-particle partitioning. Another explanation is the dynamics of the boundary layer height between day time and nighttime, which has also the effect of increasing the concentrations of these species during the night.

7) Table 1: to make this table complete, the authors may include results for HOA and COA immediately after POA.

8) Figure 2: the wind pattern can be very different between the ground level and 280 m. Therefore, I'm not sure whether it's a good idea to show the wind direction and wind speed from two different altitudes in the same panel. I suggest to show these two data measured at the same altitude (either ground or 280 m), if it's available.

9) Figure S8: in panel b), does the SMPS data start at 15 nm (as mentioned in the legend) or at 10 nm (as mentioned in the figure caption)?

**Technical comments:**

10) Page 1, line 21: while their contribution

11) Page 5, line 12: were also deployed on the roof of  another two-story building

12) Page 7, line 15: was also consistent with previous observations

13) Page 11, line 7: and also the different size distributions

14) Page 14, lines 23-25: It appears that photochemical production of less oxidized SOA was the dominant SOA formation mechanism during the control period, while aged SOA was 25 more significant after the control period

**References:**

Kuwata, M., Zorn, S. R., and Martin, S. T.: Using Elemental Ratios to Predict the Density of Organic Material Composed of Carbon, Hydrogen, and Oxygen, Environ. Sci. Technol., 46, 787-794, 10.1021/es202525q, 2012.

Sun, Y., Wang, Z., Wild, O., Xu, W., Chen, C., Fu, P., Du, W., Zhou, L., Zhang, Q., Han, T., Wang, Q., Pan, X., Zheng, H., Li, J., Guo, X., Liu, J., and Worsnop, D. R.: "APEC Blue": Secondary Aerosol Reductions from Emission Controls in Beijing, Scientific Reports, 6, 20668, 10.1038/srep20668, 2016.

Turpin, B. J., and Lim, H.-J.: Species Contributions to PM2.5 Mass Concentrations: Revisiting Common Assumptions for Estimating Organic Mass, Aerosol Sci. Technol., 35, 602-610, 10.1080/02786820119445, 2001.

---

## Author Comment (AC1) · 5 Dec 2016

**Response to Reviewers' comments**

We are thankful to the four reviewers for their thoughtful and constructive comments that help improve the manuscript substantially. We have revised the manuscript accordingly. Listed below is our point-to-point response in blue to each comment that was offered by the reviewers.

**Response to Reviewer #1**

The work of Zhao et al. presents PM1 measurement results during and after the 2015 China Victory parade event at ground level and 260m level using a HR-AMS and an ACSM. The paper was generally well written and results are valuable to the academia and government regarding air pollution control in megacities, such as Beijing. The reviewer finds a few issues, hopefully they can be well addressed before publication, as follows

Thank the reviewer for his/her positive comments.

1. As the authors have published a couple of paper regarding the HR-AMS measurement at ground site, ACSM at 260 level, and one combining results at these two heights during APEC event, and also another paper simply compared the measurement results at two heights in 2014, it might be better to put a bit more info to compare results in this work and these previous works, focusing on the differences rather than similiarities, to let readers know clearly the new findings of this work.

Thanks for the comments. Compared with our previous studies during APEC period, this study was conducted in a period with similar regional emission controls, yet substantially different meteorological conditions. As the reviewer mentioned, some conclusions are similar. For example, substantial decreases in PM mass were observed at both ground level and 260 m during APEC and the V-day Parade in this study. However, there are also many new results from this unique study. For example, during APEC, aerosol composition changed significantly at ground level, while the changes were relatively small at 260 m. However, in this study, the average compositions of $PM_1$ were similar between ground site and 260 m, and they changed similarly due to emission controls. We also observed very different secondary organic aerosols behaviors during and after control period, and investigated the vertical differences as a function of PM pollution. In addition, new measurements, e.g., particle number concentrations and black carbon, and new analysis, e.g., bivariate polar plots were performed in this study, which gave us more insights into aerosol chemistry at different heights in urban Beijing. We thank the reviewer, and we have discussed the similarities and differences between APEC and this study in the text, and also expanded more details on the unique of this study. For example, "However, due to the limited synchronous studies at ground level and high altitude in the city, our knowledge of the sources and evolutionary processes of aerosol particles, particularly in different seasons with different meteorological conditions, is far from complete."

2. Section 3.1, NR-PM1 occupied 81% of PM2.5 mass, this value is relatively higher than ones reported in Lanzhou (cite: Atmos. Chem. Phys., 14, 12593-12611), Nanjing (cite: Atmos. Chem. Phys., 16, 9109-9127,) and previous values in Beijing. It is likely that addition of BC increase the ratio, but the reviewer feels more discussion are needed. For example, does this ratio increase or decrease with the total PM2.5 mass loading? Probably, at high PM2.5 loadings, the mass fractions of supermicron meter particles increased, while at relatively clean periods, more secondarily formed species reside in submicronmeter range? It might be interesting to check.

We agree with the reviewer that adding BC certainly increases the ratio of $PM_1$ to $PM_{2.5}$. Following the reviewer's suggestions, we checked the variations of $PM_1/PM_{2.5}$ ratios as a function of PM, which is shown in Figure 1 below. As we can see that $PM_1/PM_{2.5}$ ratios vary substantially even at the similar PM levels, and we did not observe a clear ratio dependence on PM levels. One reason is because the ratio not only depends on sources and processes, but also is affected by meteorological conditions, e.g., high RH facilitates hygroscopic growth.

[Figure]

Figure 1 (a) Time series and (b) box plot of $PM_1$ to $PM_{2.5}$ ratio and $PM_{2.5}$ mass concentrations.

3. I believe the measurement uncertainties from HR-AMS and ACSM were constrained before their deployments, correct? it is not clear in the manuscript. For example, if the same air mass is loaded into these two instruments simutaneously, these two instruments should give the same concentrations for different species? correct?

Yes, the two instruments were compared at ground site side by side before their deployments. The ACSM concentrations were further corrected using the ratios from the comparisons with AMS measurements because ACSM tends to overestimate nitrate and underestimate sulfate concentrations due to the uncertainties in relative ionization efficiency and ion transmission efficiency (Budisulistiorini et al., 2014). If the same air mass is measured by the two instruments, similar concentrations for non-refractory species would be expected. Following the reviewer's suggestions, we expanded the discussions on the inter-comparisons of the two instruments in the experimental section.

4. Did the authors try to do PMF analyses individually on control period and non-control period? Although I understand the amount of data may be not enough in particular for the control period to conduct a robust PMF analyses, but it may be worth a try.

Thanks for the suggestion. We did try to do PMF analysis on control period and non-control

period separately, and some factors were not robust as we expected, e.g., HOA (low concentrations during the control period). One reason is the not enough data for the robust solution as the reviewer mentioned. In fact, the major reason we did PMF analysis on the combined period is to constrain the PMF factors to be the same during and after the control period, and then to better evaluate the effects of regional emission controls on different source factors.

**General comments:**

This manuscript reports results obtained during a field campaign undertaken at Beijing during five weeks in summer 2015. Measurements took place during and after a period of strict emission controls implemented by the Chinese Authorities to ensure good air quality in Beijing during the China victory day parade. Results clearly show an improvement of the air quality during the emission control period.

The methodology presented in this manuscript is very similar to the "APEC Blue" paper published recently by the same group (Sun et al., 2016): a high-resolution time-of-flight aerosol mass spectrometer (HR-ToF-AMS) deployed at a ground site, an aerosol chemical speciation monitor (ACSM) at 260 m on a meteorological tower, and a comparison of the particle concentration and chemical composition during and after the emission control period. However, the air quality issues faced by the inhabitants of Beijing are really impressive, so this kind of studies is of prime importance to assess the efficiency of the emission controls implemented by the Chinese Authorities. I warmly recommend the publication of this manuscript after the authors address the following comments.

Thank the reviewer for his/her positive comments.

**Specific comments:**

1) The authors used two different instruments measuring particle size distributions, i.e. the AMS and the SMPS. Each time that the authors give a particle diameter, I think it would be important to mention whether it corresponds to mobility diameter ($D_m$) or vacuum aerodynamic diameter ($D_{va}$).

Thanks for the suggestion. All diameters presented in the paper are mobility diameters ($D_m$) because we did not include the size distributions of AMS measurements which are typically reported in vacuum aerodynamic diameter ($D_{va}$).

2) Page 6, line 23: the authors used the chemical composition of the AMS to calculate the density. I'm wondering whether they also made the scatterplot of the AMS total mass vs. SMPS volume. If the size cut-off of the instruments is not the same, they can use the PToF data of the AMS, and integrate the AMS size distributions over the same size range as the SMPS.

A good comment. As shown in Fig. 2, the SMPS volume was also highly correlated with AMS $PM_1$ mass concentration. The reviewer pointed out a good way to evaluate the instruments. We didn't do such comparisons because (1) AMS measurements have a lens transmission efficiency, typically 100% for 60 – 600 nm particles (Jayne et al., 2000) , (2) the comparisons of size distributions of SMPS and AMS are largely affected by particle shape. As shown in a previous study, e.g., Zhang et al. (2005), the size distributions from SMPS measurements often have large differences from those of AMS measurements, (3) we didn't measure the

size distributions of BC and mineral dust, which are likely important fractions of $PM_1$.

[Figure]

Figure 2 (a) Time series of chemical composition dependent density, particle volume and mass concentration from SMPS and $PM_1$ mass concentration from AMS, and scatter plots of (b) SMPS volume concentrations and (c) SMPS mass concentrations versus AMS $PM_1$ mass concentrations.

3) Page 6, line 25: for the density of organic aerosols, the authors used the value given by Turpin and Lim (2001) (1.2 g/cm$^3$). However, they can also use the formula given by Kuwata et al. (2012) to calculate the density of organics by using elemental ratios (O/C and H/C).

Good point. We calculated the density of organics using elemental ratios of O/C and H/C (see Fig. 3 for the time series), and the average density of organics is 1.27 (±0.10) g/cm$^3$, which is 6% higher than 1.2 used in this study. Because the change of organic density from 1.2 to 1.3 has a minor impact on the bulk density of aerosol particles (< 3%), we kept 1.2 in the revised manuscript to be consistent with previous studies.

[Figure]

Figure 3. Time series of the density of organics which is derived from O/C and H/C ratios (Kuwata et al., 2012).

4) Page 8, lines 23-26: the authors mention that the nitrate contribution to PM$_1$ was higher at 260 m than at ground level, and suggest that this result might be due to favorable gas-particle partitioning of nitrate with low temperature at 260 m. However, according to Table 1, the difference of temperature between ground level and 260 m was less than 2°C. Is this sufficient to have a significant impact on the gas-particle partitioning of nitrate?

Thanks for the comment. The temperature in Table 1 is the average for a long period. In fact, the temperature difference between ground and 260 m was often larger than 2°C, which might play an important role in nitrate partitioning. In addition, we added another possible explanation, which is enhanced nitrate formation from heterogeneous hydrolysis of dinitrogen pentoxide at high altitude associated with high O$_3$. Now this sentence reads:" Similar vertical differences were reported in previous studies in Beijing, which were likely caused by the enhanced nighttime nitrate formation at high altitude associated with high O$_3$ (Brown and Stutz, 2012), and the favorable gas-particle partitioning of nitrate due to low T at 260 m (Sun et al., 2015a;Chen et al., 2015)"

5) Page 9, lines 19-31: there is a long discussion about the more oxidized- and less oxidized-OOA. The main problem here is that two important data are missing in this manuscript: the volatile organic compounds (VOCs) and the solar radiation. Without this information, we don't have any idea on the reason for which the MO-OOA was significantly lower during the emission control period. This could be due to the reduction of VOCs emissions (as mentioned several times by the authors) but also to reduced photochemistry.

Thanks for the comment. Unfortunately, we didn't have VOCs and solar radiation measurements in this study. Typically, more oxidized OOA is aged for a longer time in the atmosphere, and is dominantly from regional transport. Thus, the decrease in VOCs over a regional scale would cause a corresponding decrease in SOA, and hence a reduction of the amount of SOA in Beijing that is transported from outside Beijing via a long time aging. According to the O$_3$ levels in this study (Fig. 4), the O$_3$ levels during the control period appear to relatively lower than those after control period. Therefore the reduced photochemistry

might be another reason. We agree with the review that the VOCs and solar radiation data would help interpret the SOA changes in this study, which should be definitely explored in future studies.

[Figure]

Figure 4. Time series of $O_3$ at ground level.

6) Page 12, line 15: the authors mention that the diurnal patterns observed for ammonium, nitrate, and chloride were mainly due to temperature dependent gas-particle partitioning. Another explanation is the dynamics of the boundary layer height between day time and nighttime, which has also the effect of increasing the concentrations of these species during the night.

Thanks for the comment. We added such an explanation in the revised manuscript.

7) Table 1: to make this table complete, the authors may include results for HOA and COA immediately after POA.

Thanks for the suggestion. We have added HOA and COA data into Table 1.

8) Figure 2: the wind pattern can be very different between the ground level and 280 m. Therefore, I'm not sure whether it's a good idea to show the wind direction and wind speed from two different altitudes in the same panel. I suggest to show these two data measured at the same altitude (either ground or 280 m), if it's available.

Thank for the reviewer for pointing this out. Wind direction and wind speed were both available at the two heights. Following the reviewer's suggestion, we used wind direction and wind speed measured at 280 m in the revised manuscript.

9) Figure S8: in panel b), does the SMPS data start at 15 nm (as mentioned in the legend) or

at 10 nm (as mentioned in the figure caption)?

Sorry for the mistake. The SMPS measurements at 260 m starts from 15 nm. We corrected this in the revised manuscript.

**Technical comments:**

10) Page 1, line 21: while their contribution

Changed

11) Page 5, line 12: were also deployed on the roof of  another two-story building

Deleted

12) Page 7, line 15: was also consist*ent* with previous observations

Changed

13) Page 11, line 7: and also the differen*t* size distributions

Changed

14) Page 14, lines 23-25: It appears that photochem*i*cal production of less oxidized SOA was the domi*n*ant SOA formation mechanism during the control period, while aged SOA was 25 more signi*f*icant after the control period

Changed

**References:**

Kuwata, M., Zorn, S. R., and Martin, S. T.: Using Elemental Ratios to Predict the Density of Organic Material Composed of Carbon, Hydrogen, and Oxygen, Environ. Sci. Technol., 46, 787-794, 10.1021/es202525q, 2012.

Sun, Y., Wang, Z., Wild, O., Xu, W., Chen, C., Fu, P., Du, W., Zhou, L., Zhang, Q., Han, T., Wang, Q., Pan, X., Zheng, H., Li, J., Guo, X., Liu, J., and Worsnop, D. R.: "APEC Blue": Secondary Aerosol Reductions from Emission Controls in Beijing, Scientific Reports, 6, 20668, 10.1038/srep20668, 2016.

Turpin, B. J., and Lim, H.-J.: Species Contributions to PM2.5 Mass Concentrations: Revisiting Common Assumptions for Estimating Organic Mass, Aerosol Sci. Technol., 35, 602-610, 10.1080/02786820119445, 2001.

This manuscript presents a comprehensive study using a suit of on-line instruments aiming to describe the air quality improvement under the emission control and the vertical distribution of particulate matter in Beijing during the 2015 China victory day parade. The results show that the mass concentration of PM1, during and after the parade, are significant different (~50% decreased) and the chemical composition and mass concentration at ground site and 260m tower general varied synchronously, suggesting the ground site also representing a regional signal. These results are very useful for validating the strategies of emission control and evaluating the radiation forcing of PM in the boundary layer in the future. The topic in this manuscript is fitted with the range of ACP and the paper is also well written, and the results present an interest for the scientific community. This paper should be accepted on completion of the minor revisions/clarification requested below.

Thank the reviewer for his/her positive comments.

**Major comments**

1. It is interesting to compare the mass variation of each species between ground and 260m site which is useful to know the respective of ground observation. As shown in Fig. 6, all species at both sites generally display similar trends. One suggestion is that add a scatter plot in each species following the time series.

Thank the reviewer for the suggestion. We added the correlation coefficients and slopes for different aerosol species in Fig. 6 in our previous version, and also the scatter plots in supplementary (Fig. S10).

[Figure]

Figure S10. Scatter plots of $PM_1$ species measured at 260 m versus those measured at ground level.

2. The explanation for the comparison of BC between two heights is somewhat not convincible. The uncertainties of these two aethalometers were not presented in the measurement section. Which wavelength results were used for each aethalometers? Are the data corrected for shadow effect and accumulation effect?

The BC concentrations of AE22 and AE33 were both derived from 880 nm. The new model AE33 presents relatively high quality data by using a new real-time loading effect compensation algorithm based on a two parallel spot measurement of optical absorption (Drinovec et al., 2015). After the campaign, we did a three-day inter-comparison between AE22 and AE33. The results showed that the measurement of AE22 was highly correlated with those of AE33, yet with a systematic underestimation. Therefore, the BC measured by AE22 was corrected by dividing the factor of 0.72. We expanded the details on the BC measurements in the revised manuscript. "As shown in Fig. S2, BC measured by AE22 was highly correlated with that by AE33 ($r^2$ = 0.99), yet a systemic underestimation by 28% was also observed. Thus, BC measured by AE22 at ground level was further corrected by dividing a factor of 0.72 for a better comparison with that measured by AE33 at 260 m."

[Figure]

Figure 5 (a) Time series and (b) Scatter plot of BC derived from AE 22 and AE33 at 880 nm.

**Minor comments**

1. P6, L1-2: It seems that PM1 were not total neutralized based on the scatter plots between measured and predicted ammonium (slop = 0.85-0.88).

Good point. As the reviewer mentioned, aerosol particles in this study appeared not be fully neutralized, but this will not affect the default CE(=0.5) much according to the results in Middlebrook et al. (2012). Following the reviewer's comment, we revised the word "overall" as "almost" in the revised manuscript.

2. P6: The formula for calculating the density of PM1 is wrong in the denominator. It should be [NH4NO3], [(NH4)2SO4], and [NH4Cl], other than [NO3], [SO4], and [Cl].

Thank the reviewer for the comment. As the reviewer mentioned in the above comment, aerosol particles were not fully neutralized in this study. Using $[NH_4NO_3]$, $[(NH_4)_2SO_4]$, and $[NH_4Cl]$ will introduce additional mass uncertainties, e.g., more $NH_4$, for the calculation of

densities. Therefore, we used [NO$_3$], [SO$_4$], [NH$_4$], and [Cl] rather than [NH$_4$NO$_3$], [(NH$_4$)$_2$SO$_4$], and [NH$_4$Cl] to calculate the particle densities, which is also consistent with many previous studies (DeCarlo et al., 2004;Salcedo et al., 2006;Aiken et al., 2009).

3. P8, L7-9: Please add the information of the location of Tsinghua University and the instrument used for this study.

The location of Tsinghua University was now added in the revised manuscript. "Although the average PM$_1$ concentration (11.3 µg m$^{-3}$) measured by an ACSM at Tsinghua University, approximately 5 km northwest of our sampling site, during the same period is relatively lower than that in this study"

4. P8, L10: The content of this sentence is somewhat duplicated with previous sentences (P7, L27-28).

Thank you for pointing this out. We revised the sentence in L10, P8. Now it reads: "Compared to ground level, the average PM$_1$ concentration at 260 m was also lower than that measured at 260 m during the 2014 APEC summit (24.1 µg m$^{-3}$)"

5. P8, L21-23: This explanation is too general for explaining the phenomena of increased of nitrate and decreased of OA. I suggest that you can check what kinds of sources of NOx have been closed during control period and how is the change of the level of O3.

The emission controls on NO$_x$ sources mainly included restricting the number of vehicles by alternating odd and even plate numbers and shutting down factories and power plants in adjacent provinces. The NO$_x$ level was found to decrease substantially during the control period, while the O$_3$ level was comparable during and after the control period (Liu et al., 2016). Such changes in precursors plus the higher temperature and lower RH during the control period would unexpectedly suppress secondary aerosol formation.

6. P9, L13: The variation of mass concentration of COA is more than 30% higher during after control period than control period, which is not slightly.

Thank you for pointing this out. We revised this sentence as: "Although the average COA concentration during the control period (2.89 µg m$^{-3}$) was 32% lower than that after the control period (4.26 µg m$^{-3}$), the variations and concentrations of most COA peaks were similar between control and non-control periods (Fig. 3b). ".

7. P10, L10: Are the average contributions of SOA to OA at both ground sit and 260 m all 65%?

Yes. We checked the data again. The contributions are the same.

8. P11, L12-13: For the vertical variation of BC, does the uncertainty of the instruments in these

two heights account for the strange variation? Please add some information for these two Aethalometers measurement in the section 2.2. simutaneously, these two instruments should give the same concentrations for different species? correct?

Thanks for the comment. The BC measurements by AE22 and AE33 were evaluated after the campaign. As discussed in our response above, the BC measurements from the two instruments were highly correlated, and the BC from AE22 was further corrected before performing the comparisons with that from AE33. Following the reviewer's suggestions, we expanded the details on the inter-comparisons between AE22 and AE33, and also ACSM and HR-AMS in section 2.2.

This study investigates the chemical composition and potential formation pathways of aerosols at ground level and 260 m during and after the control period in Beijing. The authors provide detailed analysis for the chemical composition and evolution data. The contribution and the effect of local control and transportation are emphasized in this study, providing precious assessment for the regional emission control impact on haze treatment. This paper should be considered for the publication in ACP if the following issues are further stated.

Thank the reviewer for the positive comments.

**Major issues,**

1. In section 3.2 and section 3.3, the authors mainly talk about the difference between mass concentration of different species during and after the control period. The authors then stated the importance of emission control on air quality. However, as the main pollution level driver, the meteorological difference between two periods are merely talked about in these two part. Then, in Page 12, Line 28-29, the author mentioned that "the absence of the stagnant meteorological conditions during the control period", the difference of the meteorological background during two periods could bias or exaggerate the effect of emission control. However, the author failed to clarify the effect of emission control when excluding the meteorological differences.

Thanks for the comment. In section 3.2 and 3.3, we mainly focused on general descriptions of chemical characteristics and differences between ground level and 260 m. We totally agree with the reviewer that the effect of emission control was significantly influenced by meteorological conditions. This is also the reason we chose five polluted episodes in section 3.7, and identified two episodes (Ep2 during the control period and Ep4 after the control period) with similar meteorological conditions and footprints. The effects of emission controls on aerosol species can then be evaluated by comparing the differences between Ep2 and Ep4. A more detailed evaluation of the relative contributions of emission controls and meteorological conditions on the decreases in PM and gaseous species needs to involve modeling analysis which is beyond the scope of this study.

2. Page 9, Line 15-18, I personally disagree with the authors' statement about COA control. According to the data provided, COA was 2.89 µg m$^{-3}$ during control period averagely, which is 32% lower than that during after control period. This is actually a lot decrease. It is true that COA emission has not been overall controlled and managed, the difference of concentration between these two periods should imply the difference of accumulation process of pollutants and/or meteorological conditions.

This conclusion was drawn mainly based on the time series of COA in Fig. 3, which shows relatively similar peaks during and after the control period. The higher concentration after

the control period was mainly caused by the three large COA peaks on 18, 25, and 27 September. We agree with the reviewer that our previous statement is not accurate, and we reworded it in the revised manuscript as: "Although the average COA concentration during the control period (2.89 µg m$^{-3}$) was 32% lower than that after the control period (4.26 µg m$^{-3}$), the variations and concentrations of most COA peaks were similar between control and non-control periods (Fig. 3b)."

We also checked the three high COA peaks after the control period. They all occurred during days with relatively low PM levels. Thus, local cooking emission with favorable wind directions rather than accumulation process was likely the major cause for the higher concentration after the control period.

3. Page 9, Line 30, "regional emission controls slowed down the aging processes of OA by decreasing its precursors of volatile organic compounds"? Is there any lab/field study support for this theory?

There are two explanations for this hypothesis. One is the decreases in precursors of VOCs and NO$_x$ caused a corresponding decrease in O$_3$ (see Fig. 4 above), and hence slowed down the photochemical aging process. The second possibility is the decreases in PM levels (and hence surface areas) due to emission controls would decrease the gas-particle partitioning of semi-volatile organic species on preexisting particles. Unfortunately, we don't have lab/field study to support this hypothesis. Certainly, this is an interesting topic which should be explored in the future studies.

**Minor issues,**

Page 2, Line 3, Huang's nature and Guo's PNAS are analysis of a relatively short scale of time of air quality compared to the authors statement "air pollution during the past decade". It is obviously inappropriate to only cite these two articles for this statement.

Thanks for the comment. We have added more references here. Now it reads: "Beijing, the capital of China with ~21.71 million people in the metropolitan area (Beijing Municipal Bureau of Statistics, 2015), has been suffering from severe air pollution during the past decade (Sun et al., 2006;Chan and Yao, 2008;Sun et al., 2012;Huang et al., 2014;Guo et al., 2014;Zhang et al., 2015b;Sun et al., 2016)."

Page 6, Line 20-25, Why the chloride concentration is directly divided by the density of ammonium chloride?

AMS and ACSM only detect non-refractory chloride which mainly exists in the form of ammonium chloride. That is the reason we use the density of NH$_4$Cl, which is consistent with previous studies (Aiken et al., 2009;Salcedo et al., 2006).

Page 14, Line 23, misspell "photochemal"

Corrected

Page 21, Figure 2, what do different colours of pie charts mean?

The legend for the pie charts was added into the revised manuscript. The different colors represent different species, including organics, sulfate, nitrate, ammonium, chloride, and black carbon.

**Reference:**

Aiken, A. C., Salcedo, D., Cubison, M. J., Huffman, J. A., DeCarlo, P. F., Ulbrich, I. M., Docherty, K. S., Sueper, D., Kimmel, J. R., Worsnop, D. R., Trimborn, A., Northway, M., Stone, E. A., Schauer, J. J., Volkamer, R. M., Fortner, E., de Foy, B., Wang, J., Laskin, A., Shutthanandan, V., Zheng, J., Zhang, R., Gaffney, J., Marley, N. A., Paredes-Miranda, G., Arnott, W. P., Molina, L. T., Sosa, G., and Jimenez, J. L.: Mexico City aerosol analysis during MILAGRO using high resolution aerosol mass spectrometry at the urban supersite (T0) – Part 1: Fine particle composition and organic source apportionment, Atmos. Chem. Phys., 9, 6633-6653, 10.5194/acp-9-6633-2009, 2009.

Budisulistiorini, S. H., Canagaratna, M. R., Croteau, P. L., Baumann, K., Edgerton, E. S., Kollman, M. S., Ng, N. L., Verma, V., Shaw, S. L., Knipping, E. M., Worsnop, D. R., Jayne, J. T., Weber, R. J., and Surratt, J. D.: Intercomparison of an Aerosol Chemical Speciation Monitor (ACSM) with ambient fine aerosol measurements in downtown Atlanta, Georgia, Atmos. Meas. Tech., 7, 1929-1941, 10.5194/amt-7-1929-2014, 2014.

Chen, C., Sun, Y. L., Xu, W. Q., Du, W., Zhou, L. B., Han, T. T., Wang, Q. Q., Fu, P. Q., Wang, Z. F., Gao, Z. Q., Zhang, Q., and Worsnop, D. R.: Characteristics and sources of submicron aerosols above the urban canopy (260 m) in Beijing, China, during the 2014 APEC summit, Atmos. Chem. Phys., 15, 12879-12895, 10.5194/acp-15-12879-2015, 2015.

DeCarlo, P. F., Slowik, J. G., Worsnop, D. R., Davidovits, P., and Jimenez, J. L.: Particle Morphology and Density Characterization by Combined Mobility and Aerodynamic Diameter Measurements. Part 1: Theory, Aerosol Sci. Technol., 38, 1185-1205, 10.1080/027868290903907, 2004.

Drinovec, L., Močnik, G., Zotter, P., Prévôt, A. S. H., Ruckstuhl, C., Coz, E., Rupakheti, M., Sciare, J., Müller, T., Wiedensohler, A., and Hansen, A. D. A.: The "dual-spot" Aethalometer: an improved measurement of aerosol black carbon with real-time loading compensation, Atmos. Meas. Tech., 8, 1965-1979, 10.5194/amt-8-1965-2015, 2015.

Jayne, J. T., Leard, D. C., Zhang, X., Davidovits, P., Smith, K. A., Kolb, C. E., and Worsnop, D. R.: Development of an Aerosol Mass Spectrometer for Size and Composition Analysis of Submicron Particles, Aerosol Sci. Technol., 33, 49-70, 10.1080/027868200410840, 2000.

Kuwata, M., Zorn, S. R., and Martin, S. T.: Using elemental ratios to predict the density of organic material composed of carbon, hydrogen, and oxygen, Environ. Sci. Technol., 46, 787-794, 10.1021/es202525q, 2012.

Liu, H., Liu, C., Xie, Z., Li, Y., Huang, X., Wang, S., Xu, J., and Xie, P.: A paradox for air pollution controlling in China revealed by "APEC Blue" and "Parade Blue", Sci. Rep., 6, 34408, 10.1038/srep34408, 2016.

Middlebrook, A. M., Bahreini, R., Jimenez, J. L., and Canagaratna, M. R.: Evaluation of Composition-Dependent Collection Efficiencies for the Aerodyne Aerosol Mass Spectrometer using Field Data, Aerosol Sci. Technol., 46, 258-271, 10.1080/02786826.2011.620041, 2012.

Salcedo, D., Onasch, T. B., Dzepina, K., Canagaratna, M. R., Zhang, Q., Huffman, J. A., DeCarlo, P. F., Jayne, J. T., Mortimer, P., Worsnop, D. R., Kolb, C. E., Johnson, K. S., Zuberi, B., Marr, L. C., Volkamer, R., Molina, L. T., Molina, M. J., Cardenas, B., Bernabé, R. M., Márquez, C., Gaffney, J. S., Marley, N. A., Laskin, A., Shutthanandan, V., Xie, Y., Brune, W., Lesher, R., Shirley, T., and Jimenez, J. L.: Characterization of ambient aerosols in Mexico City during the MCMA-2003 campaign with Aerosol Mass Spectrometry: results from the CENICA Supersite, Atmos. Chem. Phys., 6, 925-946, 10.5194/acp-6-925-2006, 2006.

Sun, Y., Du, W., Wang, Q., Zhang, Q., Chen, C., Chen, Y., Chen, Z., Fu, P., Wang, Z., Gao, Z., and Worsnop, D. R.: Real-Time Characterization of Aerosol Particle Composition above the Urban Canopy in Beijing: Insights into the Interactions between the Atmospheric Boundary Layer and Aerosol Chemistry, Environ. Sci. Technol., 49, 11340-11347, 10.1021/acs.est.5b02373, 2015.

Zhang, Q., Canagaratna, M. C., Jayne, J. T., Worsnop, D. R., and Jimenez, J. L.: Time- and size-resolved chemical composition of submicron particles in Pittsburgh: Implications for aerosol sources and processes, J. Geophys. Res., 110, 10.1029/2004jd004649, 2005.